# Creating complex protocells and prototissues using simple DNA building blocks

Nishkantha Arulkumaran[1], Mervyn Singer [1], Stefan Howorka[2] & Jonathan R. Burns [2]✉

Building synthetic protocells and prototissues hinges on the formation of biomimetic skeletal frameworks. Recreating the complexity of cytoskeletal and exoskeletal fibers, with their widely varying dimensions, cellular locations and functions, represents a major material hurdle and intellectual challenge which is compounded by the additional demand of using simple building blocks to ease fabrication and control. Here we harness simplicity to create complexity by assembling structural frameworks from subunits that can support membrane-based protocells and prototissues. We show that five oligonucleotides can anneal into nanotubes or fibers whose tunable thicknesses and lengths spans four orders of magnitude. We demonstrate that the assemblies' location inside protocells is controllable to enhance their mechanical, functional and osmolar stability. Furthermore, the macrostructures can coat the outside of protocells to mimic exoskeletons and support the formation of millimeter-scale prototissues. Our strategy could be exploited in the bottom-up design of synthetic cells and tissues, to the generation of smart material devices in medicine.

Skeletal frameworks play a key role in defining biological cells and tissues[1–5]. Within eucaryotic cells, cytoskeletal microtubules, microfibers and intermediary filaments stabilize the cells' structure while facilitating dynamic morphological change, movement, and intracellular transport[6,7]. By comparison, fibers in the extracellular matrix are adhesive points that connect cells into tissues[3]. Replicating the complex fibrous structures and functions in the context of cell-like vesicles[8–12] is of basic scientific interest and can be used to generate protocells and prototissues[13–15] to study diseases[16], formulate new drugs, and generate intelligent bioactive materials[11,17–19]. To maximize their biomedical potential, these systems should be compatible with conventional drug administration routes, such as, intravenous delivery, and not adversely affect human blood cells. To achieve considerable impact, it is imperative that the biomimetic frameworks capture the complexity of the biological templates, which in turn mandates a tunable bottom-up design. Previously, biological protein fibers have been assembled inside water-in-oil droplets[20], or lipid bilayer membranes[12,21].

In place of proteins, DNA nanotechnology[22] offers an attractive route to a simpler rational design, as demonstrated by dedicated lattices on membrane surfaces[23] or ring-like scaffolds around nanoscale vesicles[24]. However, to reach the complexity of biological structural frameworks, biomimetic fibers should be tunable in dimensions and nanomechanical properties across the microscale, as well as position inside and outside of vesicles to help form prototissues which are able to display collective behavior. To simplify fabrication, these frameworks should be assembled from a minimum set of building blocks which are readily available, stable, customizable, and biocompatible with mammalian cells for possible medical applications.

We set out to address these issues by assembling complex cytoskeletal frameworks within protocells from just five oligonucleotides (Fig. 1). The component strands assemble into a DNA nanotube

[1]Bloomsbury Institute of Intensive Care Medicine, Division of Medicine, University College London, London WC1E 6BT, UK. [2]Department of Chemistry, Institute of Structural and Molecular Biology, University Collegfige London, London WC1H 0AJ, UK. ✉e-mail: Jonathan.burns@ucl.ac.uk

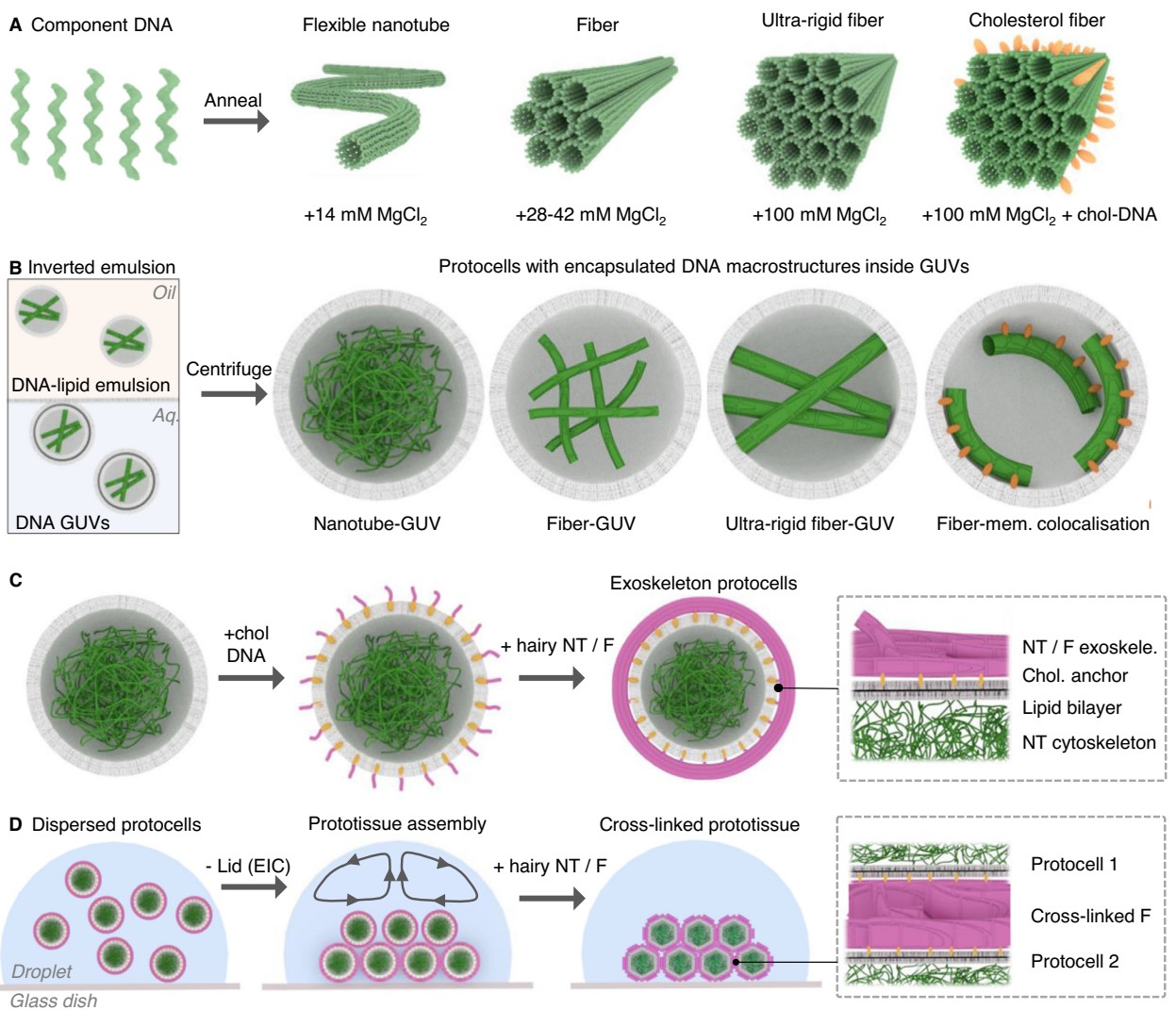

**Fig. 1 | Overview of procedure to construct DNA cyto- or exoskeletal protocells and prototissues. A** Five component oligonucleotides anneal to form DNA nanotubes (NT), or fibers (F) (green) at different MgCl₂ concentrations, DNA nanotubes form at 14 mM MgCl₂, while fibers with increasing diameter and stiffnesses form at higher Mg concentrations, left to right, and cholesterol-labeled (orange ellipsoids) 100 mM MgCl₂ DNA fibers, far right. **B** Protocells generated using an inverted emulsion procedure enable DNA macrostructures to be compartmentalized inside GUVs. DNA structures (green) in aqueous buffer are added to a lipid-oil layer (yellow box) which forms an emulsion droplet (gray sphere) surrounded by a lipid monolayer. Upon centrifugation, the dense droplets migrate to the aqueous phase (blue box), which is covered in a lipid monolayer (gray line) to form lipid bilayer protocells containing internalized DNA cytoskeletons. **C** Exoskeleton protocells are generated by adding external nanotubes or fibers (magenta) using an optional two-step procedure. First cholesterol labeled DNA single strands are added which bind to GUVs, then "hairy" DNA nanotubes or fibers are added which hybridize to coat protocells. **D** Assembling prototissues from protocells using evaporation induced convection (EIC), attractive forces and cross-linking DNA fibers (left to right). GUVs are deposited in a glucose-aqueous droplet (blue) on a glass surface (gray line), upon EIC and weak electrostatics, organize into prototissues, which can be cross-linked by adding additional "hairy" DNA nanotubes or fibers to modify prototissue morphology and connectivity. DNA nanotubes vary in diameter from 7–22 nm; for clarity the DNA nanotubes are represented as 14 helix bundles.

(NT) which was first presented by Rothemund et al.[25]. Recently, Agarwal and colleages[26] compartmentalized this DNA nanotube design inside water-in-oil droplets. Zhan and coworkers expanded on the concept and adhered vesicles or gold nanoparticles along internalized DNA frameworks[27]. In our previous work, we showed the five DNA strands could form fiber (F) condensates[28] which can offer finite control over their dimensions, stiffnesses and chemical functionality; but their potential for forming synthetic cells was not explored[25,28,29]. In this manuscript we set out to create modular cytoskeletal frameworks by self-assembling the few oligonucleotides into narrow flexible nanotubes or ultralong-wide and rigid fibers by varying the concentration of the counterion Mg²⁺ (Fig. 1A)[28]. Once folded, the macrostructures are compartmentalized inside giant unilamellar vesicles (GUVs) (Fig. 1B, left panel) with the ability to control their location using optional membrane anchors (Fig. 1B)[30,31]. In addition,

the frameworks can be placed on the outside of protocells to act as exoskeletons which can enhance the containers' mechanical stability (Fig. 1C), and to serve as a foothold for weak electrostatic interactions between protocells to form prototissues mediated via evaporation induced convection (EIC) (Fig. 1D)[32]. Complexity of the array can be increased further by adding DNA fibers which cross-link exoskeletons to fine-tune prototissue morphology, connectivity, and dynamics (Fig. 1D).

## Results

### Assembling DNA macrostructures

DNA nanotubes and fibers of tunable dimensions can be formed efficiently. To fold the superstructures, the five component oligonucleotides were dissolved in equimolar ratios at 100 μM, supplemented with magnesium chloride at different concentrations (Fig. 1A, for sequences

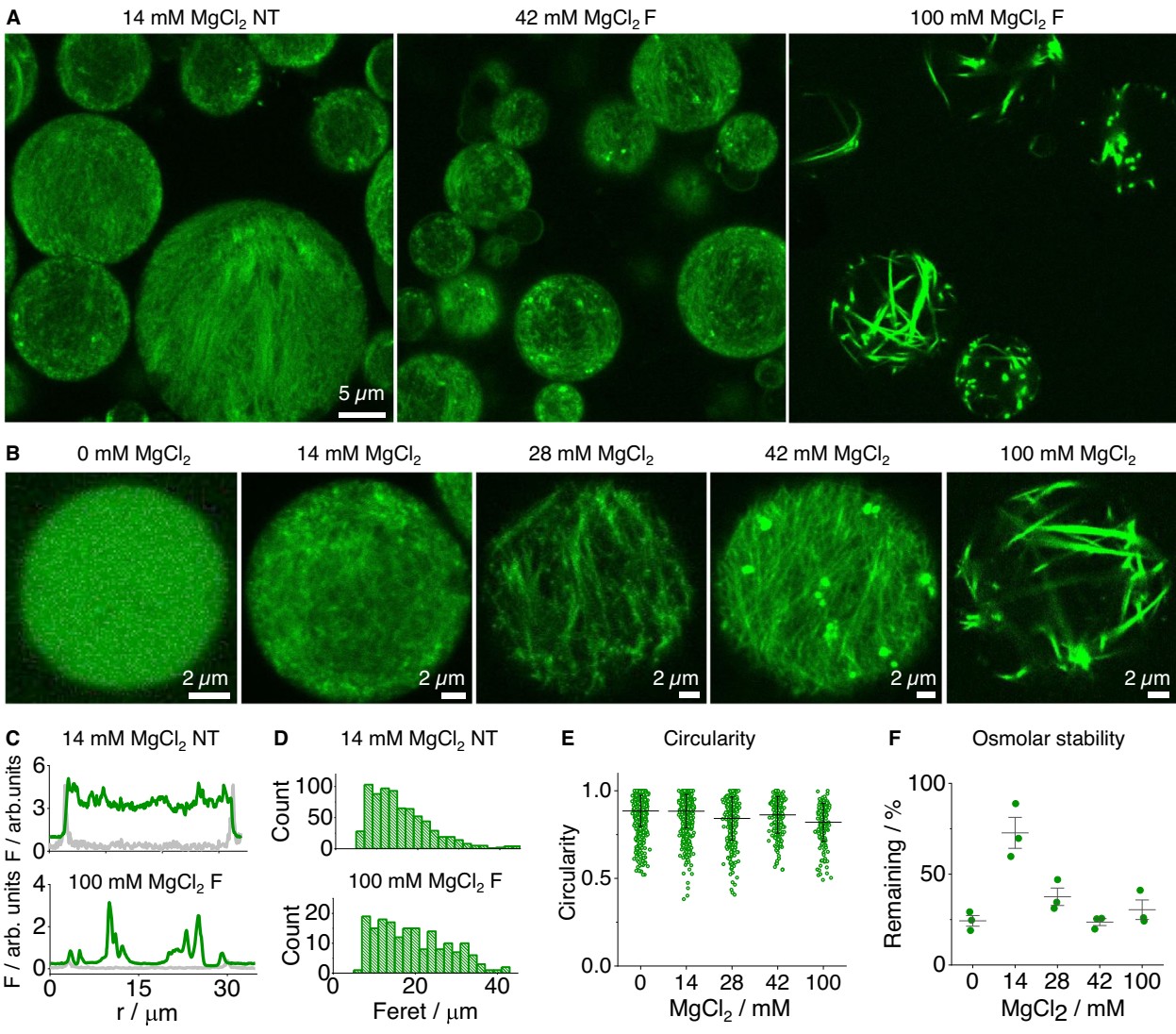

**Fig. 2 | Structural characterization of protocells. A** Confocal laser scanning microscope (CLSM) images of compartmentalized Cy3-labeled DNA nanotubes (NT) and fibers (F) inside giant unilamellar vesicles (GUVs), 14 mM MgCl₂ DNA nanotubes, and 42 or 100 mM MgCl₂ DNA fibers inside POPG GUVs, scale bar 5 μm. **B** Left to right, comparison of GUVs with encapsulated DNA nanotubes and fibers pre-folded with increasing MgCl₂ concentration, all scale bars 2 μm. **C** CLSM-derived cross section profiles of DNA nanotubes and fibers, membrane-lipid channel (gray line) and DNA channel (green line). **D** CLSM-derived diameter histogram plots of DNA nanotube and fiber protocells, $n = 747$ and 189, for NT and F, respectively. **E** Circularity dot plot of DNA nanotube and fiber protocells, $n = 914$, 747, 356, 221 and 189, left to right, respectively. Line and error bars represent the mean and standard deviation, respectively. **F** Hypertonic osmolar stability dot plot of the stated protocells remaining as a percentage at +550 mOsmol. Line and error bars represent the mean and range respectively, from 3 independent repeats.

and mixing see Tables S1–3, for 2D maps see Figs. S1–3), and then annealed by heating and cooling the mixture to form the desired frameworks within 1 h. Folding in 14 mM MgCl₂ generated DNA nanotubes, while at higher MgCl₂ concentrations (28, 42 and 100 mM MgCl₂) inter-nanotube metal complexation yield fiber condensates with wildly different dimensions and stiffnesses[28]. Alternatively, assembled DNA nanotubes can be condensed into fibers upon addition of magnesium chloride at room temperature as shown using confocal laser scanning microscopy (CLSM) (Figs. S4 and S5). Transmission electron microscopy (TEM) identified the diameter of DNA nanotubes to be 22.0 ± 4.6 nm and 42 mM MgCl₂ fibers to be 150.8 ± 75.9 nm, with the widest fibers measuring over 400 nm across (Fig. S6 and S7). Under the high oligonucleotide folding concentrations, nanotubes formed hydrogels, while fibers settled under modest centrifugal forces (Fig. S8). A centrifugal pelleting assay also established that the DNA structures formed efficiently at high ratios > 79.0 ± 6.6 % (Fig. S8). Centrifugation had no apparent detrimental effect on DNA nanotubes or fibers under these conditions (Fig. S9). By exploiting their pelleting properties, the

fiber constructs were resuspended into 14 mM MgCl₂ TAE pH 8.3 to harmonize buffer and salt conditions in downstream experiments.

## Forming protocells with different DNA cytoskeletons

DNA nanotubes and fibers were internalized into protocells using an inverted emulsion procedure (Fig. 1B)[30]. In this process, folded DNA macrostructures are mixed with a high-density sucrose solution to form dispersed water-in-oil droplets. The droplets are then added to a glucose water-layer to form a biphasic solution. Upon centrifugation, the droplets cross the phospholipid-coated oil-water phase to form giant unilamellar vesicles (GUVs) (Fig. 1B). The successful encasing of DNA nanotubes and fibers inside GUVs composed of 1-palmitoyl-2-oleoyl-sn-glycero-3-phospho-(1′-rac-glycerol) (POPG) was confirmed using confocal laser scanning microscopy (CLSM) (Figs. 2A, B and S10). Given their small nm-diameter size, DNA nanotubes exhibited fine filament textures and were, in general, homogenously distributed across the vesicles (Fig. 2A, 14 mM MgCl₂; Fig. S10). Increasing the MgCl₂ concentration resulted in longer fibers (Fig. 2A, 42 mM MgCl₂),

with the largest and most rigid fibers forming at the highest folding $Mg^{2+}$ concentration. The largest fibers were long enough to span across entire GUVs (Fig. 2A, 100 mM $MgCl_2$). The different distributions of DNA macrostructures were also tracked by comparing their fluorescence intensity profiles within protocells (Fig. 2C). In a negative control, DNA strands without $MgCl_2$ showed a homogenous distribution with no distinguishable textures (Figs. S4 and S10).

DNA cytoskeletons were also assembled inside protocells by folding the component oligonucleotides above the macrostructures melting temperature. CLSM analysis showed annealing oligonucleotides within protocells from 55 °C was high enough to generate the cytoskeleton frameworks. The temperature requirement was confirmed as folding from 45 °C, which is below their melting temperature, generated aggregate particles with heterogenous shapes (Figs. S11 and 12). The following experiments were conducted using the original fabrication method.

The encapsulated DNA nanostructures remained static within protocells as demonstrated by fluorescence-loss-in-photobleaching (FLIP) (Fig. S13). In this assay, an intense light beam was focused onto a region of Cy3-tagged DNA cytoskeleton inside a protocell, and the loss of fluorescence intensity measured over time. As exemplarily shown for 14 mM $MgCl_2$ DNA nanotubes, FLIP led to rapid bleaching and a well-defined bleached spot (Fig. S13) indicative of a static arrangement within surface-settled protocells. By comparison, non-assembled DNA strands displayed a slower bleaching profile and an overall reduction in fluorescence across the entire vesicle by 10.2% (Fig. S13), which is consistent with freely diffusing oligonucleotides.

We next determined whether the various DNA macrostructures maintained the protocells' membrane tightness, overall dimensions, roundness, and circularity. A membrane tight-seal of the protocells was confirmed by adding the small-molecule hydrophilic dye 5,6-carboxyfluorescein into the exterior solution, which was not able to permeate across the lipid bilayer membranes (Figs. S14 and 15). We then confirmed the unilamellar nature of the protocells by employing a protein nanopore alpha hemolysin (αHL). This nanopore of defined dimensions can only span single but not multiple bilayers. In the assay, the small dye Oyster 647 (O647) and the large dye green fluorescent protein (GFP) were encapsulated inside protocells. After addition of αHL at 100 nM, only O647 but not GFP displayed dye transport to the exterior (Fig. S16). These results showed size-specific transport and confirmed that the protocells are composed of unilamellar bilayers. In subsequent CLSM analysis, the protocell Feret diameter, an object's diameter along a specific direction, ranged from 5–45 μm when the cytoskeleton was composed of DNA nanotubes (Figs. 2D and S17). This was comparable to the control, oligonucleotides but without magnesium chloride (0 mM $MgCl_2$), indicating nanotubes did not detrimentally influence the vesicles during and after formation. The largest fibers displayed a slightly broader size distribution (Figs. 2D and S17). Similarly, analysis on roundness and circularity indicated that the different cytoskeleton types did not generally distort the shape of vesicles (Figs. 2E and S17, 18). A greater dependency of protocell shape was found for vesicles prepared with phospholipid 1,2-dimyristoyl-sn-glycero-3-phosphocholine (DMPC), which is widely used for drug delivery in liposomal medicine[33]. Using membranes of DMPC: cholesterol (2:1 mol ratio), non-spherical GUVs were formed for DNA fibers but not nanotubes (Fig. S19). This was most likely due to strong electrostatic interactions between DNA fibers and the charge-neutral lipid headgroups mediated by $Mg^{2+}$. However, a reduced interaction and increased spherical shape were achieved by introducing 20% mol POPG to the membrane formulation, or by reducing the $MgCl_2$ concentration of the DNA fibers from 14 to 1.4 mM prior to encapsulation (Fig. S19).

After characterizing the protocells, we explored whether the DNA cytoskeletons conferred a functional benefit in terms of stability against osmotic stress. In our assay, a hypertonic solution of 1500 mM glucose was added to the exterior of protocells containing nanotubes or fibers, which was compared to isotonic solution at 400 mM glucose. Under the ultra-high osmolarity conditions, vesicles with 14 mM $MgCl_2$ DNA nanotubes remained largely intact, while the number of remaining vesicles with other DNA fiber types displayed a significant reduction (Figs. 2F and S20). These results confirmed that DNA nanotubes had the greatest stabilization effect on the protocells. This is most likely due to their high surface and contact area with the membrane caused by the nanotubes' innate ability to form a hydrogel at high micromolar concentrations (Fig. S8). This effect was confirmed by lysing the protocells using surfactant (Fig. S21). After membrane lysis, nanotube cytoskeletons retained their shape while the biggest and stiffest fibers, which were able to curve along the protocells' membrane, changed significantly due to the polymer's high-tension state inside the protocell.

## Constructing protocells of advanced complexity

Using the custom compartmentalized DNA cytoskeletons, we set out to expand the complexity of the resulting protocells to mimic complex microtubule and actin filament properties found in eucaryotic cells[1]. This included multiple cytoskeleton types, cytoskeleton proximity location along the membrane interface, cytoskeleton real-time alignment using an external stimulus, and intracellular vesicle immobilization along cytoskeletons.

First, multiple DNA macrostructures were introduced into the same vesicle. To generate the complex cytoskeleton protocells, two types of 100 mM $MgCl_2$ DNA fibers labeled with either Cy3 or FAM were compartmentalized within the same vesicle. CLSM analysis showed both fiber types remained separate (Figs. 3A and S22). This approach was also successful for different DNA cytoskeletal types, including FAM-labeled 14 mM $MgCl_2$ DNA nanotubes and Cy3-labeled 100 mM $MgCl_2$ DNA fibers (Fig. S22). The images show heterogenous DNA macrostructures with different polymer properties can be introduced into the same protocell. These results indicate that nanotubes and fibers under the tested conditions are not in dynamic equilibrium which is unexpected given previous findings[25,34]. Therefore, additional analyses on each macrostructure combination without membranes was conducted to establish if the lack of dynamic behavior after folding was due to compartmentalization (Fig. S23). However, even in solution, the combined macrostructures did not dynamically rearrange after 48 h at room temperature, confirming our initial observations.

Second, the location of compartmentalized DNA fibers within protocells was controlled using cholesterol lipid anchors. Hydrophobic cholesterol modifications were site-specifically incorporated along 100 mM $MgCl_2$ DNA fibers to facilitate localization along the internal membrane perimeter. Cholesterol anchors were added to DNA fibers via single strand "hair" extensions (see Tables S1 and 2, and Figs. S3 for 2D map). The resulting hydrophobic macrostructures localized along the membrane giving rise to distinct halos (Fig. S24). In detailed microscopic analysis, fiber textures appeared along the membrane periphery in some GUVs (Fig. S24). Complementary FLIP analysis confirmed the integrity of the membrane-bound fibers (Fig. S24). To showcase the selectivity of the approach, we also encapsulated unmodified DNA fibers into the same protocell. The cholesterol and non-modified macrostructures were labeled with different fluorophores. Cholesterol-tagged fibers localized predominantly along the encasing lipid bilayer, while the unmodified fiber remained homogenously distributed across the vesicle interior (Figs. 3C and S25).

Thirdly, the location of compartmentalized DNA fibers within protocells was controlled in real-time using magnetic particles coupled with an external magnet. To achieve control, two-micrometer-sized magnetic beads were chemically tethered along DNA fibers using biotin-streptavidin coupling chemistry[35,36]. The biotin modifications

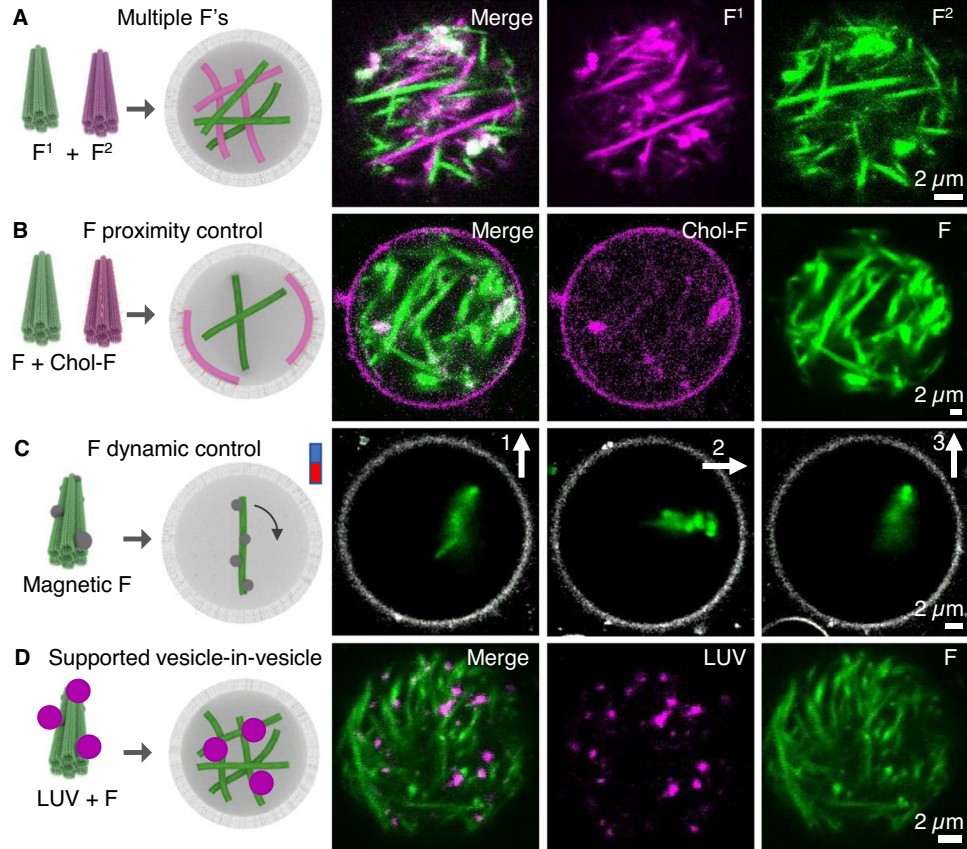

**Fig. 3 | Showcasing DNA cytoskeletal complexity and biophysical control inside protocells. A** Two fiber types compartmentalized inside GUVs, schematic representation (left), and corresponding CLSM images (right), fiber 1 (magenta) and fiber 2 (green), scale bar 2 μm. **B** Controlling location of cytoskeletal DNA fibers using cholesterol lipid anchors, schematic representation, cholesterol modifications (orange spheres) and corresponding CLSM images, cholesterol fibers (magenta) and control unmodified fibers (green), scale bar 2 μm. **C** Controlling DNA fiber alignment in real-time using magnetism, DNA fibers coated in magnetic beads (black spheres) which align with an external magnetic field inside a GUV (grey channel), arrow direction denotes the direction of the external magnetic, scale bar 2 μm. **D** Immobilization of large unilamellar vesicles (LUVs) (magenta channel) along cytoskeleton DNA fibers (green channel) inside protocells, scale bar 2 μm.

were not detrimental to fiber formation, as the modified fibers' size and shape was indistinguishable to native fibers (Fig. S26). The biotin-fibers were attached to streptavidin-coated magnetic iron particles and incorporated inside GUVs. Once inside, the iron-fibers displayed highly sensitive magnetic properties leading to real-time controllable movement by applying an external magnetic field (Figs. 3D and S27). The fibers co-aligned to an alternating parallel and perpendicular alignment of the external magnetic field (Fig. 3D).

Finally, large unilamellar vesicles (LUVs) were anchored onto cytoskeleton DNA filaments inside GUVs to mimic vesicle cargo bound via motor proteins to biological microtubules. FAM-labeled LUVs were adhered to Cy3-lableled 100 mM MgCl$_2$ DNA fibers and incorporated into protocells during the protocell formation step. The LUVs (magenta channel, Figs. 3D and S28) were successfully internalized with the cytoskeleton fibers upon GUV formation (green channel) and complexed to the fiber cytoskeleton support via favorable magnesium-ion bridging interactions. This interaction was confirmed as their position remained fixed throughout time-series CLSM analysis. This was in contrast to protocells without DNA cytoskeletons, where the LUVs location frequently changed between frames during imaging (Fig. S28).

### Assembling protocells with DNA fiber exoskeletons

Biological exoskeletal frameworks support the structure of bacteria and several eukaryotic cells[3,5]. We synthetically replicated aspects of these higher-order architectures by coating protocells with DNA fibers. The exoskeleton units were constructed by binding cholesterol lipid anchor-modified oligonucleotides to 100 mM MgCl$_2$ DNA fibers carrying "hairy" complementary strands (Figs. 1C and 4A). To facilitate fluorescence visualization of the successful assembly, exoskeleton DNA fibers, lipid anchor oligonucleotides, and cytoskeletal DNA nanotubes were tagged with three different fluorophores, Cy3, FAM, and Cy5, respectively. After assembling the protocols, CLSM images showed well-defined exoskeletal fibers wrapping around vesicles' perimeter while their lumen was filled with cytoskeletal nanotubes (Fig. 4B). Lipid membrane anchoring was required for exoskeleton formation as a construct without cholesterol-DNA exhibited 98% less fiber fluorescence at the protocell periphery (Figs. 4C and S29).

The membrane integrity of the exoskeleton-coated protocells was confirmed using a nuclease digestion assay. DNase I enzyme was added to the outside of the protocells to establish that exoskeleton fibers were accessible for digestion, while the cytoskeletal structure was shielded by the membrane container. Time series CLSM images revealed that the external fibers were digested within 2 min, while the compartmentalized cytoskeleton was unaffected (Figs. 4D and S30), thereby confirming that a tight membrane seal was maintained throughout.

Exoskeleton DNA fibers also conferred a functional benefit by stabilizing protocells against human serum which is crucial for future biomedical applications in liposomal medicine. Stability was monitored using a dye-release assay in which 5,6-carboxyfluorescein was encapsulated inside lipid-hydrated POPG vesicles at a high concentration of 200 mM. Any membrane rupturing dilutes the dye into the ambient leading to a significant increase in fluorescence emission (Fig. S31)[37]. Three vesicle types with either exoskeletal DNA fibers,

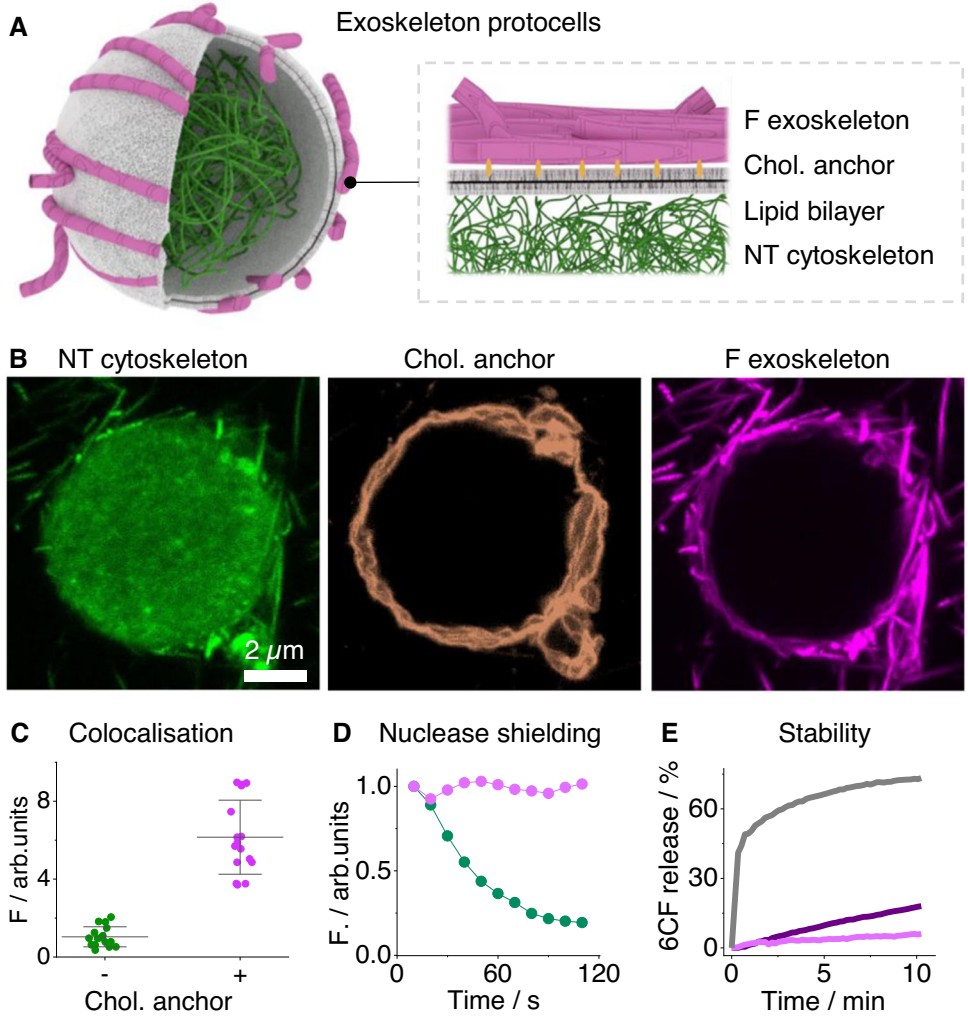

**Fig. 4 | Constructing exoskeleton protocells. A** Schematic representation. **B** CLSM images of exoskeleton protocell, nanotube (NT) cytoskeleton (green channel), cholesterol lipid anchors (orange channel) and hairy fiber (F) exoskeleton (magenta channel), scale bar 2 μm. **C** Dot plot of fiber exoskeleton fluorescence bound to GUVs either without (−) or with (+) cholesterol lipid anchors. Line and error bars represent the mean and standard deviation, respectively. **D** CLSM-derived digestion profile of cytoskeleton DNA fibers (magenta line) and exoskeleton fibers (green line) from a single protocell after the addition of DNase I. **E** 1% v/v human serum induced 6CF rupturing profiles comparing vesicles encased in either fibers (pink line), nanotubes (purple line) or unmodified (gray line). Line profiles represent the averages of three independent repeats.

exoskeletal DNA nanotubes, or without any support were assayed for their stability towards 1% v/v human serum. A fluorescence time-series revealed that exoskeleton DNA fibers stabilized the vesicles the most, while nanotubes and no-exoskeleton vesicles lysed quickly within 3 min (Fig. 4E). Human serum most likely ruptures the latter two vesicle types due to ionic imbalances and membrane disrupting proteins.

## Assembling prototissues from protocells

We developed a novel mechanism to assemble protocells into prototissues with defined mechanical properties. The higher-order networks were generated by evaporation-induced convection (EIC) (Fig. 1D). At the start of EIC, a suspension of protocells is deposited as a droplet onto a substrate surface. The droplet contains 400 mM glucose which prevents sticking of vesicles to the substrate. Exposing the suspension to air causes evaporation-induced lamellar flow which forces the vesicles to migrate to the middle of the droplet to form a prototissue[32]. CLSM analysis showed the resulting protocell arrays to be hundreds of microns across (Fig. 5A). Prototissues were formed from protocells containing both DNA cytoskeletons and exoskeletons, with each skeletal structure displaying the expected localization within the synthetic tissue (Fig. 5A, green, bottom; magenta, top). Magnified images showed

distinguishable protocells both at the edge and center of the prototissue (Fig. 5B, C). Consistent with the bottom-up design, the cytoskeleton structure occupied the core of each protocell, while the DNA exoskeleton surrounded the exterior of the protocells within the tissue network. During evaporation, the external glucose concentration increases which may potentially impact the different DNA macrostructures. We calculated the extreme high glucose condition and showed DNA fibers were not affected by the higher glucose concentrations observed during the EIC process (Fig. S32). This is line with the non-charged nature of glucose. However, if the external solution contains crowding agents, such as, PEG or divalent metals, additional checks are required to ensure nanotubes are not unintentionally condensed into fibers (Fig. S5).

The EIC-mediated mechanism of prototissue formation was visualized by particle tracking analysis. Each protocell's proceeding path (gray lines) and final location (magenta circle) were identified using CLSM time-series analysis which showed the vesicles' migration into the center of the droplet (Figs. 5E and S33). The migration of the protocells to a defined point led to weak electrostatic interactions between vesicles. This could be interrupted by adding bovine serum albumin (BSA) to mask the favorable electrostatic

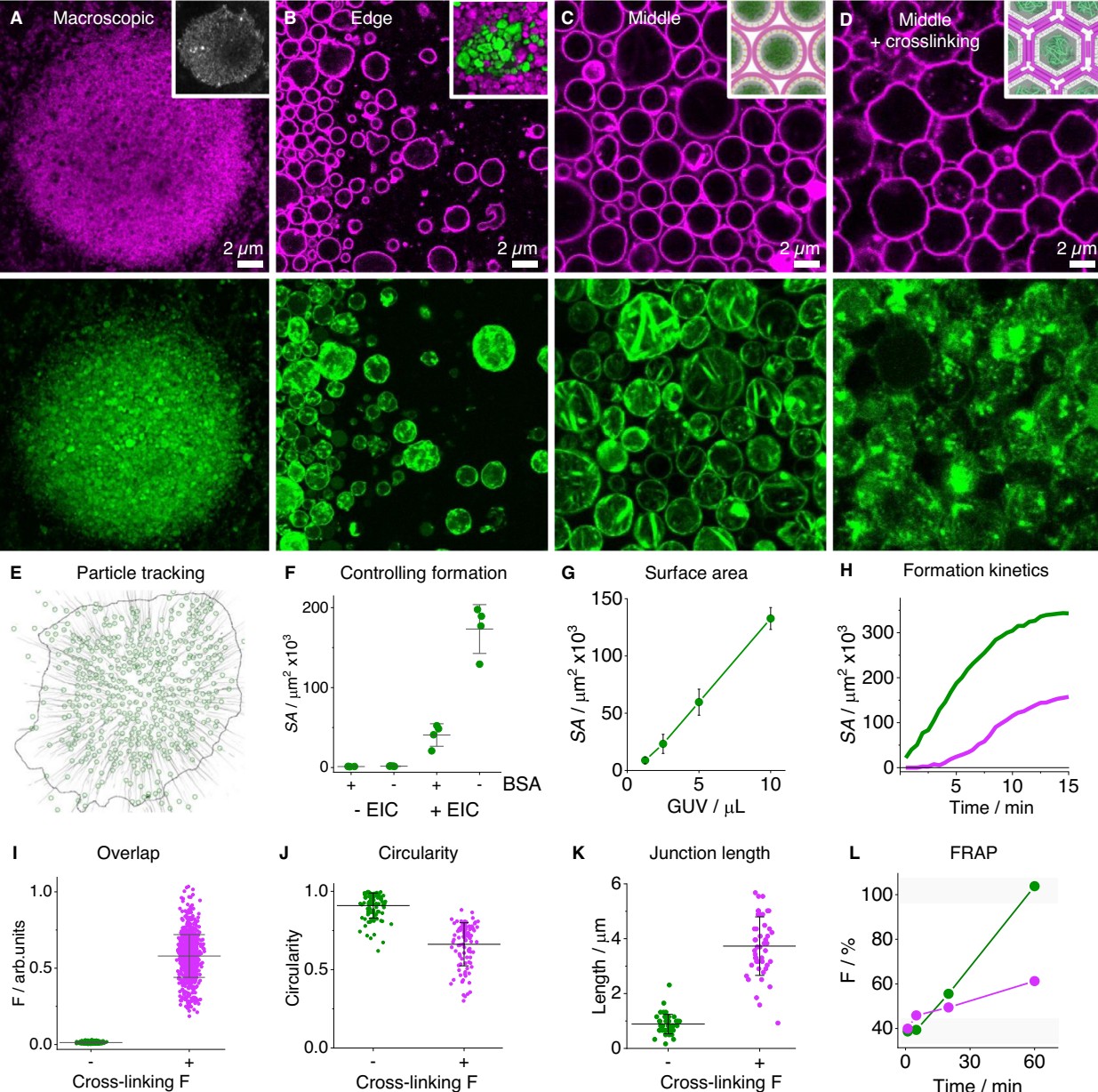

**Fig. 5 | Developing functional prototissues assembled from protocells.** CLSM images of prototissues, **A** macroscopic, **B** edge, **C** middle view, and **D** middle view of exoskeleton-DNA fiber (F) cross-linked prototissues, protocell exoskeleton (magenta channels, top row), cytoskeleton (green channels, bottom row) and insets, left to right, macroscopic image of cross-linked DNA fibers (gray), multi-layered prototissue, and schematic representations without and with cross-linking. **E** Particle tracking analysis of prototissues, end position (green circles) and tracks (gray lines) with annotated prototissue edge (black line). **F** Prototissue surface area dot plot in the absence or presence of evaporative induced convection (EIC), with and without bovine serum albumin (BSA). Line and error bars represent the mean and standard deviation, respectively, obtained from four independent repeats. **G** Plot of prototissue surface area with increasing protocell volume deposited in droplet, circles and error bars represent the mean and range respectively, obtained from three independent repeats. **H** Prototissue surface area growth over time with (magenta line) and without (green line) exoskeletons. Cross-linked prototissues showing, **I** DNA fiber co-localization sum fluorescence dot plot, line and error bars represent the mean and standard deviation, respectively, obtained from 799 measurements from single a experiment, protocell **J** circularity dot plot, obtained from 104 measurements obtained from a single experiment, **K** junction length dot plot obtained from 47 measurements, line and error bars represent the mean value and standard deviation, respectively, obtained from a single experiment, and **L** FRAP profiles comparing prototissues with and without cross-linking DNA fibers obtained from a single experiment.

interactions (Fig. 5F). Only under EIC and in the absence of BSA did prototissues form (Figs. 5F and S34). Using this insight, we expanded the concept further to construct a multi-layered prototissue. Two different protocells were added to the same droplet 30 min apart, leading to a prototissue with distinct inner and outer layers (Fig. 5B inset and Fig. S35). In a control assay, adding both protocell types simultaneously formed a homogenous prototissue array (Fig. S36).

The EIC-mediated formation was scalable as the prototissue size correlated linearly to the amount of deposited protocell (Figs. 5G and S37). The largest prototissue we were able to construct measured over 1 mm across. During formation, the prototissue followed sigmoidal kinetics which plateaued after 15 min under the assayed conditions (Figs. 5H and S38). The growth kinetics was influenced by the DNA exoskeleton; vesicles without exoskeletons formed faster and generated larger prototissues (Fig. 5H, green line). When present, the

type of exoskeleton influenced the packing density of protocells within the prototissue. With nanotubes on the protocells' exterior, an extracellular-like matrix formed between protocells, while the fiber exoskeleton generated tightly packed prototissues with reduced inter-vesicle space (Fig. S39).

Next, we studied the mechanical stability of the prototissues under high osmotic stress. This was achieved by letting the droplet volume of the assembled prototissue evaporate to half its volume, thereby increasing the osmolarity from isotonic to +400 mOsmol. The high osmotic stress led to partial collapse of the prototissues' proto-cells (Fig. S40). However, protocell deformation was reversible on returning the system back to isotonic levels by replenishing the water lost previously to evaporation. Within seconds, the isotonic conditions caused the protocells to return to their original spherical configura-tions (Fig. S40), although some cytoskeleton structures localized at the membrane periphery. These results showed the weak electrostatic interactions between protocells were stable enough to maintain inter-vesicle connectivity under high osmotic stress.

### Changing prototissue connectivity via cross-linking

To achieve prototissues with greater stability and connectivity, "hairy" DNA fibers (Figs. 1D and S3) were added to assembled prototissues to form a cross-linked tissue network. The hairy fibers contained toe-hold single strand regions which were complementary to exoskeleton DNA coatings around protocells. Upon hybridization, the external DNA fibers cross-linked the protocells to change their connectivity and morphology, as shown by microscopic analysis using Cy5-labeled cross-linking fibers (Fig. 5A inset, Figs. 5D and S41–43). The cross-linking fibers overlapped significantly with exoskeleton sites around protocells (Fig. S41–43). The binding was specific because in a control assay without any exoskeletons, the protocells showed no cross-linking fiber overlap under these conditions (Figs. 5I and S41). Upon magnification, cross-linking changed the exoskeleton protocells' shape from spherical to honeycomb-like with linear junctions (Figs. 5D and S42). In particular, the circularity of the protocells reduced from $0.9 \pm 0.1$ to $0.7 \pm 0.1$ (Fig. 5J), while the junction length between pro-tocells increased from $0.9 \pm 0.4\,\mu m$ to $3.8 \pm 1.2\,\mu m$ (Fig. 5K) upon cross-linking.

Cross-linking fibers transformed protocell dynamics within pro-totissues. The protocell mobility inside a prototissue was tracked using a fluorescence recovery after photobleaching (FRAP) assay. To con-duct the experiment, a $140 \times 140\,\mu m$ region of a prototissue was bleached, and the fluorescence recovery monitored over time (Fig. 5L and S44). FRAP analysis showed the fluorescence of prototissues without cross-linking recovered after 60 min due to dynamic rear-rangement of protocells in the network. However, in the cross-linked prototissue, the FRAP profile was 3-fold slower and only partially recovered after 60 min (Fig. S44). Finally, we showed the cross-linked prototissues could be cleaved from the dish surface using a pipettor. During the process the cross-linked prototissue fragmented into smaller mobile clusters which maintained their honeycomb-like pro-tocell connections (Fig. S45). Overall, these results revealed protocell morphology and mobility within a prototissue network can be mod-ified using external cross-linking DNA fibers to help support advanced biophysical control. By cross-linking protocells within a prototissue, the network can be cleaved from the substrate surface enabling it to be transferred into other media types for future down-stream applications.

### Establishing protocell biocompatibility with human blood cells

In order for protocells and prototissues to become promising candi-dates for intravenous biomedical applications, such as, drug delivery or cell repair, it is essential they are non-toxic to human blood cells and not immunogenic. To answer these questions, we assayed the different cytoskeletal DNA protocells in whole blood from healthy human volunteers. We applied DMPC: cholesterol protocells containing either DNA nanotube or fiber cytoskeletons, and first studied their interac-tion with human red blood cells (RBCs) in Hank's buffered saline solution (HBSS) using CLSM. After 60 min of incubation at 37 °C, the images showed both protocells and RBCs remained intact (Fig. S46). To further quantify their biocompatibility, a hemolysis assay was per-formed to measure the amount of hemoglobin released from RBCs (Fig. S46). Isolated DNA nanotubes, fibers, and empty protocells were also screened in parallel. The analysis showed negligible amounts of RBCs ruptured across all constructs; protocells caused slightly more lysis (<2%) compared to the various DNA macrostructures at $10\,\mu M$ (<0.5%). Similar results were found with human white blood cells (WBCs); CLSM images showed no change in WBC or protocell mor-phology after 1 h of incubation at 37 °C. To further characterize any response, flow cytometric analysis was performed on isolated WBCs after incubation with the stated constructs for 6 h at 37 °C. WBC via-bility was unaffected by the various protocells and DNA constructs at $1\,\mu M$. However, at $10\,\mu M$, the DNA nanotubes and fibers exhibited a 15% decrease in WBC viability (Fig. S46 and 47).

Finally, we explored if protocells elicited an immune response to human WBCs by monitoring the release of pro-inflammatory cytokine tumor necrosis factor alpha (TNF-alpha) and interleukin 6 (IL-6) (Fig. S48). Protocells and DNA constructs were incubated in healthy volunteer whole blood for 6 h at 37 °C. Enzyme-linked immunosorbent assays (ELISA) were performed which showed low doses of DNA fibers ($1\,\mu M$ oligonucleotide) and GUVs ($50\,\mu M$ lipid) elicited a minimal increase in IL-6, whereas the same concentration of DNA nanotubes was associated with a significant increase in IL-6. A similar trend was seen with TNF-alpha, higher levels of proinflammatory cytokines increased at higher DNA concentrations and protocells. Overall, our results showed both the DNA macrostructures and protocells were non-toxic to human blood cells under serum-free conditions. How-ever, further work is required to establish their full toxicity and immunogenic profiles for downstream biomedical applications. Additional steps may be taken to mitigate any immune response, for example, by coating protocells in polyethylene glycol (PEG) labeled lipids[38] and DNA structures in serum proteins[39].

## Discussion

In this report, we have utilized macromolecular DNA nanotechnology to build modular and predictable synthetic skeletal filaments inside and outside protocells and prototissues. Our findings expand the bound-aries of synthetic biology by enabling complex exo- and cytoskeletal structures to be fabricated in a simple, rapid, and predictable fashion. The cytoskeletal texture, stiffness, composition, and proximity can be custom-tailored by adding appropriate concentrations of magnesium ions, in combination with optional chemically modified oligonucleo-tides. This is in contrast to protein-based cytoskeleton analogues, which offer reduced design scope and require complex synthesis and assem-bly methods[40]. By utilizing DNA nanotechnology, our approach can be expanded to build more complex systems with advanced control, such as, multiple DNA cytoskeleton types acting constructively to mimic actin-microtubule stabilization, cytoskeleton cross-talk, or dynamic rearrangement of fibers which grow and contract on-command to induce cellular movement. Alternatively, the cytoskeletons can be decorated in synthetic motor-proteins which transport large body payloads between sites anchored to nucleus envelope mimics, or be decorated in pore-forming devices to coordinate sensing and commu-nication between inter-vesicle compartments.

Our study complements the state-of-the-art in DNA-based bio-mimetic structures. Kurokawa et al. formed the first DNA arrays inside giant unilamellar vesicles which functioned as cytoskeleton structures[41]. More recently, DNA nanotubes were compartmentalized inside water-in-oil droplets which were able to form via PEG crowding agents or transcription enzymes[26]. Using the same DNA nanotube

design, Zhan et al. were able to adhere vesicles or gold nanoparticles along DNA cytoskeleton supports[27]. In addition, Jahnke and coworkers formed cytoskeleton arrays inside lipid-bilayer vesicles using light-sensitive molecules; membrane localization was achieved by employing cholesterol lipid anchors[42]. Our work advances this field by bringing new insights into the characterization of nanotubes and fibers and their role as cytoskeleton structures inside protocells and prototissues. The repertoire of available DNA-based designer macrostructures can be easily tuned by divalent metal concentration. Once formed, the arrays are not in equilibrium, meaning multiple polymer types can be included in the same vesicle container. Multiple macrostructures can be modified with different functional chemistries to facilitate membrane co-localization, real-time dynamic morphological change, or the positioning of large unilamellar vesicles within protocells. In addition, our strategy is compatible with multiple lipid membrane types opening the door towards biomedically relevant synthetic cells and tissues.

We have exemplified the modularity of our approach by assembling protocells into millimeter-scale prototissues with unique biophysical control. Conventionally lipid-prototissues are formed using relatively harsh salt gradients or centrifugation forces[19,43]. Alternatively, protocells can be placed into position using optical tweezers[44] and printing devices[18]. However, these techniques can result in significant vesicle rupturing, deformation, or require specialist equipment and hours of preparation time. We overcame these issues by developing a fast yet gentle formation method utilizing convection and favorable weak electrostatic interactions. Our strategy is scalable and predictable across the micron and millimeter scales and enables multi-layered prototissues to be generated along 2D surfaces. Morphology and dynamics of protocells can also be studied and finetuned for tailored applications. By introducing external DNA fibers, the protocells can be cross-linked together to form honeycomb-like connections to alter their configuration, stability and protocell mobility. Complexity may be expanded in the future, for example, by introducing DNA nanopore devices anchored to external membranes which facilitate communication and collective behavior across prototissues and potentially even organoids. These developments could open the door towards smart synthetic cells and tissues used for intravenous sensing, drug delivery, and cell repair applications.

## Methods

### Ethics
The study was approved by UCL research ethics committee (REC ref 19181/001). All participants verbally consented and completed a consent form. We do not have consent to publish identifiable data. We have consent to publish healthy volunteer demographics as aggregate data or as averages. No compensation was offered to blood donors.

### Reagents
All DNA oligonucleotides were purchased from IDT DNA technologies (Coralville, IA, USA) on a 1000 nmol scale, desalted, except chemically modified DNA which was purified with HPLC and ordered on a 250 nmol scale. All other reagents were purchased from Sigma-Merck (UK) unless stated.

### Folding nanotubes and fibers
DNA nanotubes and fibers were assembled by adapting a published procedure (sequences, mixing, 2D maps and dimensions is described in the Supporting Information, see Figs. S1–S3 and Tables S1–3)[28]. The oligonucleotides were pooled in an equimolar ratio (100 μM, 1000 μL) in 1× TAE pH 8.3 with 14 mM $MgCl_2$ to form nanotubes, and 28–100 mM $MgCl_2$ to form fibers. The macrostructures were folded by heating to 95 °C for 2 min, then cooled to 60 °C at a rate of 5 °C per min, then to 20 °C at a rate of 1 °C per min using a PCR instrument (BioRad, UK). The assembled 28, 42 and 100 mM $MgCl_2$ DNA fibers were transferred into

14 mM $MgCl_2$ 1x TAE pH 8.3 (1000 μL) by centrifuging for 4 min at 14,100 × g. The supernatant (800 μL) was extracted and replaced with 14 mM $MgCl_2$ 1× TAE pH 8.3 (800 μL), the centrifugation wash-cycle was repeated two additional times. The pelleted DNA structures were resuspended by vortexing for 30 s. DNA nanotubes and fibers were stored at room temperature and used within 1 month.

### Confocal laser scanning microscopy
Images were collected using a 60× oil objective or 10× air objective CLSM (FV-1000 Olympus, UK). To visualize constructs, samples were deposited on a fluorodish (FD35-100, World Precision Instruments, Sarasota, FL, USA) and left to settle for 5 min before imaging unless stated. Microscope settings were kept identical for each experiment where appropriate. Images were analyzed using ImageJ software (https://imagej.nih.gov).

### Atomic force microscopy
The DNA nanotube solution (25 μL, 1 μM) was deposited on freshly cleaved mica, after 2 min 1× TAE 14 mM $MgCl_2$ (100 μL) was added. Images were collected using a Multimode 8 atomic force microscope (Bruker, Santa Barbara, CA, USA) in fluid tapping mode using a MSNL cantilever E tip (Bruker, Santa Barbara, CA, USA).

### Transmission electron microscopy
Carbon-coated copper grids (EM resolutions, UK) were glow discharged for 90 s (Quorum, UK). The DNA nanotubes or fibers (10 μL, 1 μM) were deposited on the grid for 1 min. The grid was then added to a uranyl formate droplet (10 μL, 2% w/v in deionized water) on parafilm for 1 s, transferred to a second uranyl formate droplet (10 μL) for 1 s, and then washed by placing on a deionized water droplet (10 μL) for 1 s. Finally, the grid was dried by blotting onto filter paper and applying a gentle airflow for 1 min. The samples were imaged using a transmission electron microscope (JEM-2100, Jeol, Tokyo, Japan) equipped with a camera (Orius SC200, Gatan Pleasanton, CA, USA). The nanotube diameter histogram profile was generated by calculating the full-width-half maximum from 37 measurements using the line section tool in ImageJ software (https://imagej.nih.gov). The widest fiber diameter histogram profile was obtained using the same procedure from 50 measurements.

### DNA nanotube and fiber pelleting assay
The stated DNA macrostructures (10 μM, 100 μL, 1× TAE 0, 14, 28, 42 or 100 mM $MgCl_2$) were pelleted in a plastic vial (0.2 mL, Eppendorf, UK) by centrifuging for 30 s at 14,100 × g. The degree of formation was determined by comparing the baseline corrected absorbance at 260 nm before and after centrifugation, then the percentage DNA in each pellet determined.

### DNA cytoskeletal protocell synthesis
1-palmitoyl-2-oleoyl-sn-glycero-3-phospho-(1′-rac-glycerol) (POPG) giant unilamellar vesicles (GUVs) were prepared by modifying a published protocol[45]. POPG (150 μL, 10 mM) and Cy5 lipid PE (0.2 μL, 1 mg per mL, where stated) in chloroform was added to a glass vial (1 mL), and the solvent removed under vacuum using a rotary evaporator for 5 min. The thin film generated was resuspended in mineral oil (150 μL, M5904, lot number MKBX0231V) by vortexing and sonicating for 1 min. The stated DNA constructs (15 μL, 100 μM, 1× TAE pH 8.3, 14 mM $MgCl_2$) were mixed with sucrose (15 μL, 725 mM) and then transferred carefully to the mineral oil layer. The suspension was vortexed for 1 min at room temperature, then carefully added to the top of a glucose solution (1 mL, 400 mM) in a plastic vial (1 mL). The vesicles were generated by centrifuging at 14,100 × g for 30 s. The mineral oil top layer and the majority of the sucrose layer (850 μL) were carefully removed. The remaining solution containing the pelleted vesicles (100 μL) was gently mixed with a pipettor, then transferred to a clean

plastic vial. During protocell formation the DNA constructs are diluted 1:1 (v/v) into the sucrose solution, therefore the final tile concentration was assumed to be 50 μM. Vesicles were gently shaken before use and used within 48 h. To generate DMPC/cholesterol/POPG protocells, the above protocol was followed, except DMPC (100 μL, 36.9 mM, in chloroform), cholesterol (50 μL, 36.9 mM, in chloroform) and POPG (20 μL, 10 mM, in chloroform) lipids were used.

For diameter, roundness, circularity and osmolality assays, green fluorescent protein (GFP) (5 μL, 50 μM, in PBS) was added to the DNA-sucrose mixture and the above protocol followed. The stated protocells (5 μL) were deposited on the fluorodish and left to settle for 5 min before imaging unless stated. For osmolarity assays, the protocells were deposited on the CLSM slide, then after 5 min additional glucose was added (5 μL, 400 mM or 1500 mM) and the remaining protocells imaged. Protocell properties were analyzed using ImageJ software (https://imagej.nih.gov) using the GFP channel. To extract the data, first, a background subtraction was performed using a 50 pixel rolling ball radius, then a threshold was applied, and the particles analyzed using the analyze particle tool with diameters set to 10 micron$^2$-infinity.

### Complex protocell synthesis

To generate multi-DNA macrostructure protocells, the protocol for encapsulation above was followed, except both FAM and Cy3-labeled fibers (7.5 μL each, 100 μM, 1× TAE pH 8.3) were mixed with sucrose (15 μL, 725 mM) and then transferred to the mineral oil layer. To generate mixed cholesterol DNA fibers and DNA fibers protocells, the protocol for encapsulation above was followed, except FAM-labeled cholesterol fibers (7.5 μL, 100 μM) and Cy3-labeled fibers (7.5 μL, 100 μM, 1× TAE pH 8.3) were mixed with sucrose (15 μL, 725 mM) and then transferred to the mineral oil layer. To generate magnetic DNA fiber protocells, the protocol for encapsulation above was followed, except biotin and Cy3-labeled fibers (15 μL, 100 μM, 1× TAE pH 8.3) were mixed with sucrose (15 μL, 725 mM) and then transferred to the mineral oil layer. The DNA fibers (50 μL, 100 μM, 1× TAE pH 8.3) were pre-coated in Dynabeads Streptavidin 65801D (ThermoFisher Scientific, UK) (10 μL) at room temperature for 30 min shaking at 1000 r.p.m. To generate vesicle-in-vesicle protocells, the LUVs were generated by adding DMPC (1000 μL, 36.9 mM, in chloroform) and cholesterol (500 μL, 36.9 mM, in chloroform) to a glass vial (14 mL) and the solvent removed under vacuum using a rotary evaporator for 20 min at 40 °C. PBS (1000 μL) was added to the film and the vial vortexed for 2 min to generate GUVs. The vesicles were washed by transferring to a plastic vial (1.5 mL), then centrifuging for 30 s, the supernatant discarded (750 μL) and replenished with fresh PBS (750 μL). The pelleting-washing step was repeated two more times. The vesicles were then extruded to diameters of 2 μm using a mini-extruder system equipped with a 2 μm filter by extruding them 15 times (Avanti Polar Lipids, AL, USA). 1 Chol single strand DNA containing a FAM dye (10 μL, 100 μM in water) was added and the solution mixed for 5 min at room temperature. The cholesterol single strand-LUVs (10 μL) were then added to fibers (10 μL, 100 μM, 1× TAE 14 mM MgCl$_2$) and sucrose (20 μL, 725 mM) and the above protocell formation protocol followed. In the control the DNA solution was replaced with buffer (10 μL, 1× TAE 14 mM MgCl$_2$).

### Exoskeleton protocell synthesis

Cy5-labeled 14 mM MgCl$_2$ DNA nanotube cytoskeleton DMPC/cholesterol protocells (60 μL) were added to 1 Chol strand containing a FAM fluorophore (5 μL, 25 μM, in water) and left to incubate at 37 °C for 15 min. Next, either hairy DNA nanotubes or hairy fibers (1 μL, 100 μM, in 14 mM MgCl$_2$, 1× TAE pH 8.3) were added and the solution left for 15 min. The exoskeleton GUVs (2 μL) were then deposited into PBS (10 μL) and imaged using CLSM. For the nuclease digestion assay, DNase I (1 μL, 1 mg per mL) (New England Biolabs, UK) in 1×

DNase I buffer was carefully added to the top of the droplet and a time series performed, scanning every 10 s. Digestion profiles for cyto- and exoskeletons were identified by plotting the fluorescence intensities over time, profiles were obtained from a single time series.

### Exoskeleton protocell human serum dye release assay

POPG (1000 μL, 10 mM) in chloroform was added to a round bottom flask (5 mL) and the solvent removed using a rotary evaporator for 20 min. 5,6-carboxyfluorescein (1 mL, 200 mM, in 1× TAE pH 8.3, 300 mM KCl) was added to the film and the solution sonicated for 30 s. Non-encapsulated dye was removed using a pre-equilibrated NAP-25 column by loading the vesicles (250 μL) and eluting with 1× TAE pH 8.3, 500 mM KCl (250 μL fractions, total volume 3 mL). The fractions containing purified vesicles were pooled. The vesicles (150 μL) were added to three plastic vials and either buffer (2 μL, 1× TAE 14 mM MgCl$_2$), cholesterol DNA nanotubes (2 μL, 100 μM in 1× TAE 14 mM MgCl$_2$) or cholesterol DNA fibers (2 μL, 100 μM in 1× TAE 14 mM MgCl$_2$) were added and left to incubate for 15 min at room temperature. Next, the various vesicle-exoskeleton constructs (22 μL) were diluted in 1× TAE pH 8.3, 500 mM KCl (178 μL) and transferred to a fluorescence cuvette and the 6CF emission monitored using a fluorometer (excitation 495 nm, emission 515 nm, PMT voltage 500, slit width 5 mm, scanning every 2 s). After 30 s, human serum (2 μL) was added and the rate of dye release plotted as a percentage before and after rupturing.

### Tissue networks

DMPC/cholesterol/POPG protocells containing either cytoskeleton Cy3-labeled nanotubes or fibers (2.5–10 μL in 400 mM glucose, with or without BSA final concentration 1 mg/mL) were deposited on a fluorodish and left to settle with or without the lid on where stated. CLSM time-series were collected across the whole, in the center, or at the edge of the synthetic tissue over the stated time frames. To generate tissue networks with exoskeleton protocells, the above procedure was followed, except FAM-labeled cholesterol fibers (1 μL, 10 μM in PBS) were added to the GUV solution (10 μL) and then deposited on the surface. To cross-link prototissues, Cy5-labeled hairy DNA fibers (2.5 μL, 10 μM in PBS) were carefully deposited onto the tissue network. For controls, PBS only was added, or to protocells without any exoskeleton cholesterol fibers. For the hypertonic assay, the prototissue droplet (10 μL) was left to evaporate to approximately half its volume, then water (5 μL) was carefully added to the top of the droplet to rehydrate the prototissue whilst performing a time series. Prototissue surface area was calculated using ImageJ particle analysis tool (https://imagej.net/software/fiji/). To extract the data, first, a background subtraction was performed using a 50 pixel rolling ball radius, then a threshold was applied and the particles analyzed using the analyze particle tool with diameters set to 10 microns$^2$-infinity. Protocell tracking during prototissue formation was established using TrackMate software embedded in ImageJ[46]. Each protocell path was identified by applying a LoG detector and simple LAP tracker using an estimated object diameter of 3 microns. To measure cross-linking fiber fluorescence, the average fluorescence intensities of 799 measurements was applied comparing with and without cross-linking obtained using the line section tool in ImageJ software. To measure protocell circularity and junction length in prototissues, the segmented line tool was used to map each protocell's membrane. To conduct FRAP assays, after prototissue formation and cross-linking, a region of interest was selectively bleached by increasing the laser intensity in the FAM channel for 30 s, then a time series collected at the stated time points, the bleached region of interest fluorescence intensity was plotted as a percentage against a non-bleached region.

## Human blood collection

Venipuncture was performed and whole blood (5–10 mL) was drawn into SST™ II Advanced Plus, EDTA (Ethylenediamine tetra-acetic acid), vacutainer (Becton Dickinson (BD) UK, Oxford, UK) from four consenting donors. For assessment of RBC stability, the whole blood (10 μL) was diluted into Hank's buffered saline solution (HBSS) (1 mL, Gibco, UK). To isolate WBCs, RBCs were lysed using 1× red cell lysis buffer (BD; Beckton Dickinson biosciences, UK). Cells were washed using centrifugation and re-suspended in HBSS (5 mL). All healthy volunteer blood donors were male with a median age of 36 (33–37) years old.

## CLSM analysis of protocells with human blood cells

Protocells (5 μL, 400 mM glucose) were deposited on to a fluorodish. After 5 min, PBS (10 μL) then either dilute whole blood (4 μL) or WBCs (4 μL) were added and images collected after 60 min incubation at 37 °C.

## RBC viability

The protocells (10 μL in 400 mM glucose) or DNA constructs (10 μL, 10 μM, 1× TAE 14 mM MgCl$_2$) were added to dilute human whole blood (1 μL in HBSS) and the samples incubated 60 min at 37 °C. Next, the different samples were centrifuged at $14,100 \times g$ for 1 min, and the supernatant isolated. The amount of lysed RBCs was determined by scanning the absorbance at 540 nm. Values were determined using positive (deionized water) and negative (PBS only) controls.

## WBC viability

Whole blood (5 mL) was collected in heparinized syringes from each healthy donor. Isolated white blood cells (WBCs) were used to assess viability using flow cytometry. To isolate WBCs, red blood cells (RBCs) were lysed using 1x red cell lysis buffer (BD; Beckton Dickinson biosciences, UK). Cells were washed and re-suspended in 5 mL Hank's buffered saline solution (HBSS). The protocells or DNA constructs (at 1 μM or 10 μM) were added to isolated white blood cells and the samples incubated for 6 h at 37 °C. Cell viability stain (Live/Dead; Thermo Fisher Scientific, UK) was added to cells for 30 min at 37 °C and samples assessed using flow cytometry. Heat (65 °C for 30 min) was used to kill cells as a positive control and for gating all samples. Cells were analyzed using flow cytometry on a LSR Fortessa (BD) flow cytometer (BD Biosciences) running BD FACSDiva version 9 software. Flow cytometry data were analyzed using FlowJo version 10.0 (Tree Star Inc, USA). Graphs were constructed, and statistical analysis performed using Prism version 9 (GraphPad, (San Diego, USA). Minimum of 5000 events/sample within the granulocyte population (main WBC population) were read. Data was collected from two individual experiments. Identical gates were applied to all samples. Gating strategies are shown in Fig. S47.

## Whole blood stimulation

Whole blood stimulation was used to assess inflammation. Whole blood (15 mL) was collected in EDTA vacutainers (BD) from four healthy donors. Immune cell pro-inflammatory cytokine release was determined by the addition of the protocells, DNA (single strand, nanotubes or fibers) or GUVs for 6 h. DNA was added to achieve final concentrations of either 1 μM or 10 μM. GUVs were added to achieve a final upper DMPC lipid concentration of 50 μM or 500 μM. PBS and buffer were added as extra controls. All ex-vivo cell experiments were carried out at 37 °C in a cell culture chamber. Following incubation, cells were separated from plasma by centrifugation for 10 min at $12,000 \times g$ using a refrigerated centrifuge at 4 °C. The resulting supernatant containing plasma was extracted and stored at −20 °C prior to cytokine analysis using ELISA. ELISA was used to quantify levels of proinflammatory cytokines interleukin-6 (IL-6) and tumor necrosis factor-alpha (TNF-alpha) released in whole blood assays. DuoSet ELISA kits (R&D Systems, Minneapolis, USA) were used to assess cytokine levels according to the manufacturers' instructions. Absorbance was read at 450 nm using a spectrophotometric ELISA plate reader (Anthos HTII; Anthos Labtec, Salzburg, Austria). Statistical data was analyzed using a non-parametric Kruskal–Wallis test using GraphPad Prism 5.00 (San Diego, USA).

## Statistics and reproducibility

Dot plots, box plots and graphs were generated using OriginPro (2022) or GraphPad Prism (5.00) software where stated. Figure 2a, b was repeated 10 times with similar outcomes. Figure 3a–d was repeated 5 times with similar outcomes. Figure 4b was repeated 3 times with similar outcomes. Figure 5a–c was repeated 10 times with similar outcomes. Figure 5d was repeated 3 times with similar outcomes. Figure S4 was repeated 10 times with similar outcomes. Figure S5 was repeated 5 times with similar outcomes. Figure S6a, b was repeated 5 times with similar outcomes. Figure S7a, b was repeated 3 times with similar outcomes. Figure S9 was repeated 3 times with similar outcomes. Figure S10 was repeated 10 times with similar outcomes. Figure S12 was repeated 3 times with similar outcomes. Figure S13a, b was repeated 3 times with similar outcomes. Figure S14 was repeated 3 times with similar outcomes. Figure S15a was repeated 3 times with similar outcomes. Figure S16a was conducted once. Figure S18a was repeated 3 times with similar outcomes. Figure S19a, b was repeated 3 times with similar outcomes. Figure S20a, b was repeated 3 times with similar outcomes. Figure S21a was repeated 3 times with similar outcomes. Figure S22 was repeated 5 times with similar outcomes. Figure S23a, b was repeated 5 times with similar outcomes. Figure S24a, b was repeated 3 times with similar outcomes. Figure S25 was repeated 5 times with similar outcomes. Figure S26 was repeated 3 times with similar outcomes. Figure S27 was repeated 5 times with similar outcomes. Figure S28a, b was repeated 5 times with similar outcomes. Figure S29 was repeated 5 times with similar outcomes. Figure S30a was repeated 3 times with similar outcomes. Figure S32 was repeated 3 times with similar outcomes. Figure S33a was repeated 5 times with similar outcomes. Figure S34 was repeated 3 times with similar outcomes. Figure S35 was repeated 3 times with similar outcomes. Figure S36 was repeated 3 times with similar outcomes. Figure S37 was repeated 3 times with similar outcomes. Figure S38 was repeated 3 times with similar outcomes. Figure S39a was repeated 3 times with similar outcomes. Figure S40 was repeated 3 times with similar outcomes. Figure S41 was repeated 3 times with similar outcomes. Figure S42 was repeated 3 times with similar outcomes. Figure S43 was repeated 3 times with similar outcomes. Figure S44 was repeated 3 times with similar outcomes. Figure S45 was repeated 3 times with similar outcomes. Figure S46a, b was repeated 3 times with similar outcomes.

## Reporting summary

Further information on research design is available in the Nature Portfolio Reporting Summary linked to this article.

# Data availability

All relevant data supporting the key findings of this study are available within the article and its Supplementary Information files or from the corresponding author upon reasonable request. Source data are provided with this paper.

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

## Acknowledgements

Dr. Alan Greg at the Centre for Cell and Molecular Dynamics, Division of Biosciences, Rockefeller Building, UCL. This research was funded by the Rosetrees Trust (grant M924). N.A. acknowledges salary support from the UK Medical Research Council (Grant: MR/W030489/1).

## Author contributions

Research designed by J.R.B., experiments conducted by N.A. and J.R.B., manuscript written by J.R.B. with contributions from N.A., M.S. and S.H.

## Competing interests

The authors declare no competing interests.
