## [Peer Review File · Nature Communications]

REVIEWER COMMENTS

Reviewer #1 (Remarks to the Author):

The manuscript by Arulkumaran et al describes the use of DNA nanotubes and fibers to introduce bottom-up cytoskeletons and exoskeletons (or perhaps cell wall mimics) in vesicle-based protocells and prototissues. The cytoskeletons offer enhanced stability, can be localized and manipulated inside the protocells by functional handles and magnetic fields, and provide a potential transport pathway for small vesicles. The exoskeletons enable the cohesion of protocells inside a prototissue and can be crosslinked to create stable tissue mimics with tight junctions. The use of DNA nanotubes and fibers helps to provide a good biocompatibility with for example human blood, and minimizes the immune response.

The manuscript is very well written and the experiments are well documented and carefully conducted, including an extensive number of controls. The findings are interesting for researchers from various backgrounds, including synthetic cells, DNA nanotechnology, synthetic biology and biomaterials. They also go beyond previous articles in which DNA nanostructures have been used as artificial cytoskeletal elements: here, the authors combine control over localization with transport functionality, the ability to form prototissues via exoskeleton-driven attachment and crosslinking, and compatibility with biological fluids, cells and tissues. I therefore recommend publication of this manuscript if the authors could address a few minor issues.

1. Vesicles: the authors report that they use GUVs, however proof that the vesicles are actually unilamellar when they are formed in the presence of high concentrations of DNA nanotubes and magnesium is missing. It is known that the inverted emulsion method is sensitive to parameters such as ionic strength and viscosity of the aqueous phases.

2. Protocells: the authors refer to their GUVs as protocells in the title and results, but seem to aim at “the bottom-up design of synthetic cells and tissues, to the generation of smart material devices in medicine” in their abstract and introduction. This may be a semantic discussion, but in the experience of the reviewer protocell is usually used to indicate a cell-like structure within the context of origin of life, while the authors seem to aim at cell-like structures that can mimic or interact with contemporary living cells. Would it be better to call these synthetic cells and tissues, in line with other literature in the field?

3. References: the references do not accurately reflect the significant amount of work done in the field of reconstitution of cytoskeletal structures (not only DNA based) inside lipid vesicles and studying their

effect on stability and shape of vesicles. There is interesting work by the groups of Schwille (also including inverted emulsion method), Koenderink and Ganzinger, Bausch and others. Specifically regarding DNA-based cytoskeletons, the authors should include a reference to the recent work of Jahnke et al (ACS Nano 2022, 16, 7233), and regarding DNA-based prototissues to the work of Samanta et al. (Nature Comm. 2022, 3968). I would also like to encourage the authors to discuss work by other groups on using DNA nanostructures as cytoskeletal elements in vesicles (refs. 27-29 and Jahnke et al. ACS Nano 2022) in more detail and highlight the differences with the current system.

4. Reproducibility: details on the number of repeats and reproducibility is missing for some experiments, in particular microscopic observations. E.g., Fig. 3, Fig. 4D, E, Fig. 5A-D, E, H, I, L.

5. Mixed fibers: the experiments with mixed fiber types (Fig. S19) formed at different magnesium concentrations suggests that the fibers are unable to reorganize once formed, and are therefore not equilibrium structures. The same holds for all other experiments in which fibers formed at high magnesium are resuspended in a 14 mM MgCl₂ buffer. Do the authors observe any (slow) morphological changes over time? How long after formation were the images acquired? Please comment on the arrested fiber state in the manuscript or SI.

6. Figure S4: the nanofibers seem to be aligned in a particular direction. Is this an artefact of the preparation method? Please comment.

7. Figure S5 and S6: the authors compared AFM-based diameters for nanotubes with TEM-based diameters for nanofibers (also in main text p.5). Please use the same technique for direct comparison, or state explicitly that the measurements were based on different techniques in the main text. Additionally, the authors presumably report a mean and standard deviation in the main text, which suggests that the diameters have a Gaussian distribution. However, in the case of nanofibers, this seems to be not very accurate.

8. Electrostatic interactions: on p.7 and 13 the authors suggest that electrostatic interactions between vesicles and between DNA fibers and membranes plays a role, however, with the surfaces having the same charge or a net neutral charge, this seems unlikely. Do the authors mean a different type of interaction (magnesium ion bridging, Vanderwaals, ...)?

9. Evaporation induced convection: during evaporation the concentration of magnesium and other salts increases significantly and this may affect the interaction between DNA fibers. Please comment.

10. Typo. p.3: this criteria

Reviewer #2 (Remarks to the Author):

The paper describes various experiments demonstrating the assembly of DNA fibers inside and outside double emulsion vesicles, illustrating the versatility of these structures toward the formation of multi-droplet tissues and testing their compatibility with the presence of biological cells and blood serum. Overall the manuscript is well put together with attractive figures and images, and it is appealing to a broad audience interested in the advancement of synthetic cells. The paper contributes potentially useful methods, but their description needs to be improved.

First, I am surprised that the authors didn't disclose that their tile is identical to Rothmund's SEs tile (JACS 2004, Ref 24), which in turn is adapted from the DAE-E tile proposed by Seeman in the nineties. A non-expert reader may believe these tiles are an invention of the authors because they cite their own previous work in key design-related sentences (Ref 25). The authors must clearly acknowledge their design was taken "as is" from Rothmund's paper - admitting this will not negatively impact the relevance of the work (but not saying it certainly does and is a disservice to the readers). The authors should also relate their findings to the abundant literature on DNA nanotubes in the discussion - the existence of many studies on this type of structure is a positive aspect, because the results presented here may be extended to many other nanotube types.

I find the authors characterization of nanotube fiber formation to be interesting and it got me thinking a lot. But how can the authors exclude the formation of nanotubes with very large diameter, rather than fibers composed of dozens of thin nanotubes? Since the Mg^{++} is changed prior to annealing, in my opinion this possibility can't be excluded. Have the authors observed a transition from nanotubes to fibers by changing the buffer conditions *at constant (room) temperature*?

Rothmund JACS 2004 did report nanotubes/ribbons with 100 nm+ circumference (although this was due to a change in tile design, not to a change in Mg^{++} concentration).

The melting curves at 100 mM Mg^{++} appear indistinguishable from those of DNA nanotubes at 12 mM (see data in Rothmund et al and related papers), which does not help clarify the picture, and is kind of surprising.

Assuming nanotube fibers do form, the data reported (Fig. S6 C) show their diameter varies a lot. In contrast, the drawings suggest very regular honeycomb fiber patterning and they are a bit misleading in this sense - a reader might mistake these tile-based fibers for a DNA origami architecture.

Using 100 μ M tile is 100 times the typical concentration used for annealing the SEs tiles, and in fact the authors state that a hydrogel forms at 100 μ M (Fig. S7). Is it correct to say that the addition of Mg⁺⁺ makes a hydrogel switch to well defined fibers?

I was confused by this sentence “The pelleting properties of the fiber constructs were exploited and resuspended into 14 mM MgCl₂ TAE pH 8.3 to harmonize their buffer and salt conditions in subsequent experiments.”

Does this mean all samples, regardless of annealing conditions, were pelleted and then resuspended to a solution of TAE 14 mM Mg prior to encapsulation in vesicles? So what is the effective estimated tile concentration in vesicles? And Mg⁺⁺ concentration?

What part does centrifugation play? Do we estimate any loss of DNA during the vesicle production process?

What is the tile concentration in Figs S5 (the AFM imaging concentration can't be 100 μ M here) and S6?

Fig. S4 is organized in a confusing way, and I am confused because the concentration is so high I would not expect to be able to discern nanotubes, so I don't know whether the 14 mM Mg⁺⁺ sample can be classified as “bundled”, or it's simply too crowded to see anything.

The authors must specify the timing of their experiments, meaning how long were nanotubes allowed to grow in vesicles prior to imaging. Nanotubes dynamically polymerize/depolymerize and join post annealing (Ekani-Nkodo, A., Kumar, A. & Fygenson, D. K. Joining and scission in the self-assembly of nanotubes from DNA tiles. *Phys. Rev. Lett.* 93, 268301 (2004), not cited), and this dynamic process changes their length distribution and organization. It is therefore important to specify how long they were left in the vesicles prior to imaging.

I wonder if, given enough time inside vesicles, we could observe nanotube fibers under all Mg⁺⁺ conditions. Agarwal et al. 2020, using essentially the same tile type adopted here, reports that nanotube bundles (qualitatively similar to those reported in this paper) form several hours after encapsulation in w/o droplets at 12 mM Mg⁺⁺.

There is no description of image processing methods/criteria and statistics are not always clearly discussed. The authors mention they used ImageJ, but they should specify how droplets/nanotubes were detected, counted, and tracked. How did they measure circularity etc (by hand? through a segmentation method?). What is circularity vs roundness and how are they computed? Also what is the “junction length” discussed in Fig. 5K? How were the FRAP profiles in Fig. 5 L obtained? Are we looking at a normalized average over how many droplets?

Fig. 3A: I assume the nanotubes were annealed separately with distinct Mg⁺⁺ concentration, and then mixed and encapsulated. But why did the different nanotube types not interact since they have the

same sticky ends? Were these images taken right after encapsulation? What happens after several hours? I suspect the two populations will join and mix over time - see also Ekani-Nkodo, A., Kumar, A. & Fygenon, D. K. Joining and scission in the self-assembly of nanotubes from DNA tiles. Phys. Rev. Lett. 93, 268301 (2004)

Minor:

Introduction: Refs 27 and 29 focus on DNA nanotubes in water in oil droplets, not vesicles. The Goepfrich group has done work on nanotubes in vesicles in a different paper, which should be cited (Jahnke, K., Huth, V., Mersdorf, U., Liu, N. & Göpfrich, K. Bottom-Up Assembly of Synthetic Cells with a DNA Cytoskeleton. ACS Nano 16, 7233–7241 (2022).)

The authors may want to note that the assembly of distinct DNA nanotube types (different sequences) was demonstrated in Agarwal et al. (ref 27), and SUV localization was demonstrated by Zhan et al (ref 29) albeit in water in oil droplets.

Please provide a brief explanation of what Feret diameter is.

I found lot of typos and captions that are poorly phrased or incorrect in the SI, and this leaves readers with a bad impression.

A few examples: caption of Fig. S15 C “roundness interval plots for the stated DNA macrostructure type” I thought Mg⁺⁺ is varied but the structure is the same. Caption of Fig. S17 - *ferret* diameter. Caption of Fig. S19 refers to nanotubes and fibers inconsistently when compared with the labels on the images. SI section 2.4 Melting *charaterisitcs*...

Reviewer #3 (Remarks to the Author):

The manuscript by Arulkumaran et al. reconstitutes DNA-based filaments in and around GUVs. DNA filaments have been reconstituted inside GUVs as well as outside GUVs, just not both. So in large parts, the manuscript lacks novelty. The most interesting and novel aspect is the formation of GUV “tissues”, but this is only a very small part. So the manuscript certainly has interesting elements, which deserve deeper exploration. Instead, at the moment it is a collection of somehow disconnected parts (e.g. the blood cell part in the end comes entirely out of the blue). Each of them remains superficial and does not

bring new conceptual insights. I feel like the entire paper could be reworked into a much stronger story or condensed for publication at a lower level.

Comments:

1) Fig. 3B: Do the authors show somewhere that there are actually still filaments forming along the membrane? To me it looks like its basically covering the whole membrane and that filamentous structures are no longer visible. So maybe they did not stay intact, I don't see and FRAP measurements or z-stacks showing that the DNA stays in one place/that there are really filaments as are shown in their sketch.

2) The assembly of protocell tissues is the most novel aspect; especially with the two-layer prototissue assembly. It would be great to see them use that and explore it more, e.g. how stiff is the tissue? Can the stiffness be changed? Could one attach filaments to the inside of the GUVs and then form the tissue; does that change how the internal scaffold behaves? Is there a difference in stability between GUVs in the middle of the tissue and the outside? Is the tissue actually intact (so, can it be lifted?)

3) The part with the blood cells does not seem to fit in. Why it is important for the rest of the story? Why blood cells in particular? GUVs and cells have been integrated before.

4) It would be interesting to see if the GUVs could be made to interact with the cells, for example via the external fibres, or since most of the work is on stability and the prototissue, whether the cells interact/can interact or even be incorporated into the prototissue somehow. Other than it feels like this part is from a different story/should only be a side note and not at the end of the story here.

5) Small comment: Several figures are not referenced correctly in the main text.

Reviewer #1 (Remarks to the Author):

The manuscript by Arulkumaran et al describes the use of DNA nanotubes and fibers to introduce bottom-up cytoskeletons and exoskeletons (or perhaps cell wall mimics) in vesicle-based protocells and prototissues. The cytoskeletons offer enhanced stability, can be localized and manipulated inside the protocells by functional handles and magnetic fields, and provide a potential transport pathway for small vesicles. The exoskeletons enable the cohesion of protocells inside a prototissue and can be crosslinked to create stable tissue mimics with tight junctions. The use of DNA nanotubes and fibers helps to provide a good biocompatibility with for example human blood, and minimizes the immune response.

The manuscript is very well written and the experiments are well documented and carefully conducted, including an extensive number of controls. The findings are interesting for researchers from various backgrounds, including synthetic cells, DNA nanotechnology, synthetic biology and biomaterials. They also go beyond previous articles in which DNA nanostructures have been used as artificial cytoskeletal elements: here, the authors combine control over localization with transport functionality, the ability to form prototissues via exoskeleton-driven attachment and crosslinking, and compatibility with biological fluids, cells and tissues. I therefore recommend publication of this manuscript if the authors could address a few minor issues.

We thank the reviewer for providing these constructive comments.

1. Vesicles: the authors report that they use GUVs, however proof that the vesicles are actually unilamellar when they are formed in the presence of high concentrations of DNA nanotubes and magnesium is missing. It is known that the inverted emulsion method is sensitive to parameters such as ionic strength and viscosity of the aqueous phases.

We performed an additional experiment using a protein nanopore to confirm GUVs are composed of single lipid bilayers and are not multi-lamellar (Figure S16). The protein nanopore, alpha haemolysin (α HL) has defined dimensions which can only span single but not multiple bilayers. In the assay the small dye Oyster 647 (O647) and the large dye green fluorescent protein (GFP) were encapsulated inside protocells. After addition of α HL at 100 nM, only O647, but not GFP, displayed dye transport to the exterior. These results showed size-specific transport to confirm the protocells are composed of unilamellar bilayers.

2. Protocells: the authors refer to their GUVs as protocells in the title and results, but seem to aim at “the bottom-up design of synthetic cells and tissues, to the generation of smart material devices in medicine” in their abstract and introduction. This may be a semantic discussion, but in the experience of the reviewer protocell is usually used to indicate a cell-like structure within the context of origin of life, while the authors seem to aim at cell-like structures that can mimic or interact with contemporary living cells. Would it be better to call these synthetic cells and tissues, in line with other literature in the field?

We are attempting to build ever-more complex cell mimics with DNA nanotechnology and would like to stick to this naming system. The term protocells is frequently used in the literature to describe synthetic cell-like structures which mimic nature (references <https://www.nature.com/articles/s41467-022-31632-6>, <https://www.nature.com/articles/nmat5005>, <https://www.nature.com/articles/s41467-021-23850-1>, <https://www.nature.com/articles/s41565-019-0399-9>).

3. References: the references do not accurately reflect the significant amount of work done in the field of reconstitution of cytoskeletal structures (not only DNA based) inside lipid vesicles and studying their effect on stability and shape of vesicles. There is interesting work by the groups of Schwille (also including inverted emulsion method), Koenderink and Ganzinger, Bausch and others. Specifically regarding DNA-based cytoskeletons, the authors should include a reference to the recent work of Jahnke et al (ACS Nano 2022, 16, 7233), and regarding DNA-based prototissues to the work of Samanta et al. (Nature Comm. 2022, 3968). I would also like to encourage the authors to discuss work by other groups on using DNA nanostructures as cytoskeletal elements in vesicles (refs. 27-29 and Jahnke et al. ACS Nano 2022) in more detail and highlight the differences with the current system.

Thank you for providing these references most of them have been added to the discussion section of the manuscript.

The table below outlines the overlap and differences between the stated manuscripts. A high level outline has now been added to the main text.

This paper	Jahnke 2022	Zhan et al 2022	Agarwal 2021
Three lipid used formulations with chemistry insight	One lipid composition	Water-in-oil droplets	Water-in oil-droplets
Characterisation of DNA nanotubes and fibers outside of protocells at high concentrations using magnesium	Light triggered reversible assembly of cytoskeletons	Single cytoskeleton type	Nanotube isothermal formation over time via with PEG
Revealed DNA nanotubes and fibers after folding are not in dynamic equilibrium	PEG crowding agents induced fiber formation	Reversible assembly and ATP triggered polymerisation or invader strand	Study nanotube container properties
Demonstrated unilamellar nature of GUV using protein nanopore flux	Location control of fibers along membranes	Vesicle or gold nanoparticle guided transport	DNA-RNA nanotube formation inside compartments
Customisable DNA cytoskeleton properties including chemical functionality (cholesterols) and stiffness	Characterisation of DNA nanotubes and fibers outside of protocells low concentrations	Characterisation of DNA nanotubes and fibers outside of protocells low concentrations	In situ transcription triggered formation of DNA-RNA nanotubes
Formation of cytoskeleton fibers inside protocells via thermal annealing			Enzyme mediated assembly and disassembly of DNA-RNA nanotubes
Multiple cytoskeleton types inside protocells (nanotubes + fibers, fibers + fibers, cholesterol fibers + fibers)			
Dynamic real time control of cytoskeleton fibers using magnetic particles			
Exoskeleton nanotubes or fibers around protocells			
Novel mechanism of prototissue assembly			
Prototissues with cross-linking DNA fibers			
Multi-domain prototissues			
Protocell dynamics within prototissues			
Cross-linked prototissues can be cleaved from the substrate			

4. Reproducibility: details on the number of repeats and reproducibility is missing for some experiments, in particular microscopic observations. E.g., Fig. 3, Fig. 4D, E, Fig. 5A-D, E, H, I, L.

These details have now been included in the supporting information and the figures can captions have been updated accordingly.

5. Mixed fibers: the experiments with mixed fiber types (Fig. S19) formed at different magnesium concentrations suggests that the fibers are unable to reorganize once formed, and are therefore not equilibrium structures. The same holds for all other experiments in which fibers formed at high magnesium are resuspended in a 14 mM MgCl₂ buffer. Do the authors observe any (slow) morphological changes over time? How long after formation were the images acquired? Please comment on the arrested fiber state in the manuscript or SI.

Both DNA nanotubes and fibers, under these conditions, do not reorganise and are not in dynamic equilibrium after folding. Indeed, they are very stable in solution for months at room temperature. CLSM images were collected 2-48 hours after protocell formation. A statement has been added to the materials and methods section. To further prove this point, we conducted a new assay in which two types of fibers, one labelled with a FAM dye and the other labelled with a Cy3 dye were mixed together in equimolar ratios in solution. CLSM images after 1 min and 48 hours confirmed the mixed fibers types are not in equilibrium under these conditions. These images have been added to the manuscript (Figure S23).

6. Figure S4: the nanofibers seem to be aligned in a particular direction. Is this an artefact of the preparation method? Please comment.

The fibers can align on the surface of the dish due to lamellar flow resulting from the deposition process. This effect was exploited previously in a separate manuscript (Burns J.R. Small, 2021). In addition, the DNA macrostructures can align above the surface of the dish due to their high concentration. A similar effect has been reported for protein cytoskeleton analogues (<https://www.nature.com/articles/nature11591>). In the revised manuscript, a statement has been added to the caption of Figure S4 to help clarify the situation.

7. Figure S5 and S6: the authors compared AFM-based diameters for nanotubes with TEM-based diameters for nanofibers (also in main text p.5). Please use the same technique for direct comparison, or state explicitly that the measurements were based on different techniques in the main text. Additionally, the authors presumably report a mean and standard deviation in the main text, which suggests that the diameters have a Gaussian distribution. However, in the case of nanofibers, this seems to be not very accurate.

Thank you for pointing this out. We used ImageJ processing software to measure fiber diameter distribution and the largest diameter along the fiber. We specify these details in the revised methods section and SI accordingly. Previously we also mis-labeled Figure S6 as AFM, when it fact it was a direct comparison between TEM measurements. The Figure has now been corrected.

8. Electrostatic interactions: on p.7 and 13 the authors suggest that electrostatic interactions between vesicles and between DNA fibers and membranes plays a role, however, with the surfaces having the same charge or a net neutral charge, this seems unlikely. Do the authors mean a different type of interaction (magnesium ion bridging, Vanderwaals, ...)?

Thank you - yes, we did mean magnesium-ion bridging. The text has been amended accordingly.

9. Evaporation induced convection: during evaporation the concentration of magnesium and other salts increases significantly and this may affect the interaction between DNA fibers. Please comment.

We calculated the extreme high-salt scenario and conducted an additional experiment which showed the fibers are not affected by the increased concentrations of MgCl₂ and glucose during the evaporation induced convection process. CLSM image S34 shows the DNA fibers in either 14 mM MgCl₂ 1x TAE or 2.8 mM in 800 mM glucose (upper limit when the droplet volume is halved), which showed no discernible differences.

[Calculations. 30 μL of aqueous liquid in the emulsion phase contains 7 mM MgCl_2 . The protocells end up being resuspended into 150 μL of glucose 400 mM. Therefore if all the protocells were to rupture, then the maximum final concentration of MgCl_2 would be 1.4 mM and glucose 400 mM, and if the droplet volume is halved during EIC, then the concentrations would increase to 2.8 mM MgCl_2 and glucose 800 mM]

10. Typo. p.3: this criteria

Now corrected.

Reviewer #2 (Remarks to the Author):

The paper describes various experiments demonstrating the assembly of DNA fibers inside and outside double emulsion vesicles, illustrating the versatility of these structures toward the formation of multi-droplet tissues and testing their compatibility with the presence of biological cells and blood serum. Overall the manuscript is well put together with attractive figures and images, and it is appealing to a broad audience interested in the advancement of synthetic cells. The paper contributes potentially useful methods, but their description needs to be improved.

We thank the reviewer for providing these constructive comments.

First, I am surprised that the authors didn't disclose that their tile is identical to Rothmund's SEs tile (JACS 2004, Ref 24), which in turn is adapted from the DAE-E tile proposed by Seeman in the nineties. A non-expert reader may believe these tiles are an invention of the authors because they cite their own previous work in key design-related sentences (Ref 25). The authors must clearly acknowledge their design was taken "as is" from Rothmund's paper - admitting this will not negatively impact the relevance of the work (but not saying it certainly does and is a disservice to the readers). The authors should also relate their findings to the abundant literature on DNA nanotubes in the discussion - the existence of many studies on this type of structure is a positive aspect, because the results presented here may be extended to many other nanotube types.

We apologise for the lack of clarity on the original design and have amended in the revised manuscript the introduction to make this clearer. We also cite additional DNA nanotube reviews and selected papers within the revised introduction.

I find the authors characterization of nanotube fiber formation to be interesting and it got me thinking a lot. But how can the authors exclude the formation of nanotubes with very large diameter, rather than fibers composed of dozens of thin nanotubes? Since the Mg⁺⁺ is changed prior to annealing, in my opinion this possibility can't be excluded. Have the authors observed a transition from nanotubes to fibers by changing the buffer conditions *at constant (room) temperature*?

The DNA nanotube design, at 14 mM MgCl₂ or below, can form diameters between 7-22 nm. This was identified first by Rothmund and colleagues in their seminal JACS 2004 paper using AFM, and reproduced by others, including ourselves in our recent paper (Burns, Small 2021). In this paper, we characterised the nanotubes and fibers across a range of magnesium chloride concentrations on AFM, CLSM and TEM. We showed fibers start forming at magnesium concentrations above 14 mM MgCl₂ (Small paper, Figure S9). Detailed TEM analysis revealed the fibers are composed of discrete bundles of nanotubes and not ultra-wide single nanotubes (Small paper, Figure S12-S13).

To further back up the presence of nanotube bundles, we have additionally conducted experiments for the revised manuscript to show that DNA nanotubes can be condensed into fibers at room temperature by adding higher concentrations of MgCl₂. Figure S5 shows a CLSM time-series of nanotubes at 25 μM being condensed at room temperature by adding 100 mM MgCl₂ within 126 seconds. The images show poorly defined nanotubes bundles at 0 seconds, but well-defined fibers after 126 seconds. To showcase the divalent metal requirement, we took either pre-folded DNA nanotubes or DNA fibers, then added 100 mM KCl, 100 mM MgCl₂ or 100 mM CaCl₂ at room temperature. CLSM images showed magnesium and calcium ions were able to condense DNA nanotubes into fibers, but not monovalent potassium ions (Figure S5). Pre-folded DNA fibers were unaffected by the addition of all types of additional metal under these conditions.

Rothmund JACS 2004 did report nanotubes/ribbons with 100 nm+ circumference (although this was due to a change in tile design, not to a change in Mg⁺⁺ concentration).

This is a separate DNA nanotube design and one that we have not studied. We agree other designs may be able to form larger diameters.

The melting curves at 100 Mm Mg⁺⁺ appear indistinguishable from those of DNA nanotubes at 12 Mm (see data in Rothmund et al and related papers), which does not help clarify the picture, and is kind of surprising.

The melting temperature of fibers in this paper were conducted in 50 % v/v 1x TAE 42 mM MgCl₂ 50 % v/v 725 mM sucrose. This will undoubtedly alter the melting transition compared to the Rothmund paper. However, in our 2021 Small paper we conducted a direct nanotube vs fiber comparison and observed a difference in the melting profiles. Nanotubes displayed two transitions at 47.1 and 62.0°C, whereas fibers displayed a single transition at 61.1°C. This result confirmed a high salt stabilisation effect for fibers.

Assuming nanotube fibers do form, the data reported (Fig. S6 C) show their diameter varies a lot. In contrast, the drawings suggest very regular honeycomb fiber patterning and they are a bit misleading in this sense - a reader might mistake these tile-based fibers for a DNA origami architecture.

We have added a statement in the figure legends and SI stating the nanotubes vary in diameter between 7-22 nm.

Using 100µM tile is 100 times the typical concentration used for annealing the SEs tiles, and in fact the authors state that a hydrogel forms at 100µM (Fig. S7). Is it correct to say that the addition of Mg⁺⁺ makes a hydrogel switch to well defined fibers?

Only the nanotubes form hydrogels under the assayed conditions after folding. After centrifugation all fiber constructs (28-100 mM MgCl₂) were able to pellet whilst the nanotubes at 100 µM could not. As we have discussed above, the folded nanotubes can alternatively be condensed into fibers without folding by adding magnesium chloride at room temperature.

I was confused by this sentence "The pelleting properties of the fiber constructs were exploited and resuspended into 14 mM MgCl₂ TAE pH 8.3 to harmonize their buffer and salt conditions in subsequent experiments." Does this mean all samples, regardless of annealing conditions, were pelleted and then resuspended to a solution of TAE 14 mM Mg prior to encapsulation in vesicles? So what is the effective estimated tile concentration in vesicles? And Mg⁺⁺ concentration?

Yes, this is correct. The final concentration of fibers was assumed to be quantitative due to the high folding and pelleting values observed in the centrifugation-pelleting assay (>79 %). The samples are diluted 1:1 (v/v) in sucrose solution, therefore the final concentration was assumed to be 50 µM inside the GUVs. To clarify this, a statement has been added to the Materials and Methods section of the revised manuscript.

What part does centrifugation play?

Centrifugation plays no effect on the nanotubes or fibers. CLSM images before and after centrifugation show the nanotubes and fibers after 3x centrifugation-resuspension cycles. A new Figure has been added to the SI (Figure S9).

Do we estimate any loss of DNA during the vesicle production process?

As discussed above, only minor levels of DNA macrostructure are lost either due to poor assembly (i.e. not big enough to pellet) or loss during handling. Provided the salt conditions are suitable, then the majority of the DNA macrostructure are internalised inside GUVs, this is further backed up because we see no or very little non-encapsulated DNA macrostructures in solution (see Figures S10, S14, S19).

What is the tile concentration in Figs S5 (the AFM imaging concentration can't be 100µM here) and S6?

The tile concentration for Figures S6 and S7 was 1 µM. These values have been added to the SI.

Fig. S4 is organized in a confusing way, and I am confused because the concentration is so high I would not expect to be able to discern nanotubes, so I don't know whether the 14 mM Mg⁺⁺ sample can be classified as "bundled", or it's simply too crowded to see anything.

Figure S4 was added to show what the different macrostructures look at high folding conditions to enable the reader to compare against the protocell form. To show the nanotubes are composed of bundles we have adapted the figure and show nanotubes folded at 100 µM and then diluted to 10 µM. Under these conditions DNA nanotubes at 10 µM form clear bundles.

The authors must specify the timing of their experiments, meaning how long were nanotubes allowed to grow in vesicles prior to imaging. Nanotubes dynamically polymerize/depolymerize and join post annealing (Ekani-Nkodo, A., Kumar, A. & Fygenson, D. K. Joining and scission in the self-assembly of nanotubes from DNA tiles. *Phys. Rev. Lett.* 93, 268301 (2004), not cited), and this dynamic process changes their length distribution and organization. It is therefore important to specify how long they were left in the vesicles prior to imaging.

The fibers and nanotubes, under these conditions, do not reorganise and are not in dynamic equilibrium. Indeed, they are very stable in solution for months at room temperature. To further show this, we monitored the dynamic rearrangement of NT and F, or F and F combinations outside protocells after 1 min and 48 hours (Figure S23). The results showed -as expected for us- neither NT or F in any combination dynamically rearranged.

In the main text, the mixed NT and F, or F and F combinations inside protocells were collected after 24 hours of formation, and they also showed no dynamic rearrangement.

CLSM images were collected within 2 days of protocell formation; a statement has been added to the Materials and Methods section for clarity.

I wonder if, given enough time inside vesicles, we could observe nanotube fibers under all Mg⁺⁺ conditions. Agarwal et al. 2020, using essentially the same tile type adopted here, reports that nanotube bundles (qualitatively similar to those reported in this paper) form several hours after encapsulation in w/o droplets at 12 mM Mg⁺⁺.

As stated above, we do not observe nanotubes condensing into fibers after longer periods of time in the low salt condition, either in solution or inside protocells.

There is no description of image processing methods/criteria and statistics are not always clearly discussed. The authors mention they used ImageJ, but they should specify how droplets/nanotubes were detected, counted, and tracked. How did they measure circularity etc (by hand? through a segmentation method?). What is circularity vs roundness and how are they computed? Also what is the "junction length" discussed in Fig. 5K? How were the FRAP profiles in Fig. 5 L obtained? Are we looking at a normalized average over how many droplets?

The relevant figure captions have been modified to state the number of repeats and types of measurements. We have also included descriptions for the data extraction methods in the Materials and Methods sections.

Fig. 3A: I assume the nanotubes were annealed separately with distinct Mg⁺⁺ concentration, and then mixed and encapsulated. But why did the different nanotube types not interact since they have the same sticky ends? Were these images taken right after encapsulation? What happens after several hours? I suspect the two populations will join and mix over time - see also Ekani-Nkodo, A., Kumar, A. & Fygenson, D. K. Joining and scission in the self-assembly of nanotubes from DNA tiles. *Phys. Rev. Lett.* 93, 268301 (2004)

These images were collected either on the same day and 48 hours after GUV encapsulation, and they showed no differences. Under our conditions the nanotubes and fibers do not dynamically rearrange (Figure S23).

Minor:

Introduction: Refs 27 and 29 focus on DNA nanotubes in water in oil droplets, not vesicles. The Goepfrich group has done work on nanotubes in vesicles in a different paper, which should be cited (Jahnke, K., Huth, V., Mersdorf, U., Liu, N. & Göpfrich, K. Bottom-Up Assembly of Synthetic Cells with a DNA Cytoskeleton. *ACS Nano* 16, 7233–7241 (2022).

We have included this publication as reference.

The authors may want to note that the assembly of distinct DNA nanotube types (different sequences) was demonstrated in Agarwal et al. (ref 27), and SUV localization was demonstrated by Zhan et al (ref 29) albeit in water in oil droplets.

These references have also been added to the manuscript.

Please provide a brief explanation of what Feret diameter is.

We have added a description of the Feret diameter to the main text.

I found lot of typos and captions that are poorly phrased or incorrect in the SI, and this leaves readers with a bad impression. A few examples: caption of Fig. S15 C “roundness interval plots for the stated DNA macrostructure type” I thought Mg⁺⁺ is varied but the structure is the same. Caption of Fig. S17 - *ferret* diameter. Caption of Fig. S19 refers to nanotubes and fibers inconsistently when compared with the labels on the images. SI section 2.4 Melting *characteristics*...

We apologies for these errors and have corrected them and refined the captions to make them clearer.

Reviewer #3 (Remarks to the Author):

The manuscript by Arulkumaran et al. reconstitutes DNA-based filaments in and around GUVs. DNA filaments have been reconstituted inside GUVs as well as outside GUVs, just not both. So in large parts, the manuscript lacks novelty. The most interesting and novel aspect is the formation of GUV “tissues”, but this is only a very small part. So the manuscript certainly has interesting elements, which deserve deeper exploration. Instead, at the moment it is a collection of somehow disconnected parts (e.g. the blood cell part in the end comes entirely out of the blue). Each of them remains superficial and does not bring new conceptual insights. I feel like the entire paper could be reworked into a much stronger story or condensed for publication at a lower level.

We thank the reviewer for their comments. However, we disagree with a number of points, as detailed below.

- **Novelty.** *Our manuscript offers a large degree of novelty as demonstrated in the table below. Our manuscript brings together all parts of the bottom-up assembly of protocells and prototissues with detailed chemical insight, full characterisation of each component at each step, plus expands the limits of the current-state-of-art.*
- **Biocompatibility.** *We are interested in generating biocompatible protocells and prototissues for disease detection and treatment, this means they must be non-toxic to human blood cells and must not generate significant immune response. To help clarify our requirement we have edited the introduction.*
- **Story.** *The remit is already very big, and increasing the dataset even further will not make the story line any easier to understand.*
- **Others perspective.** *Referees 1 and 2 commented positively about the content and story line of the manuscript.*

The table outlines the overlap and differences between the stated manuscripts.

This paper	Jahnke 2022	Zhan et al 2022	Agarwal 2021
Three lipid used formulations with chemistry insight	One lipid composition	Water-in-oil droplets	Water-in oil-droplets
Characterisation of DNA nanotubes and fibers outside of protocells at high concentrations using magnesium	Light triggered reversible assembly of cytoskeletons	Single cytoskeleton type	Nanotube isothermal formation over time via with PEG
Revealed DNA nanotubes and fibers after folding are not in dynamic equilibrium	PEG crowding agents induced fiber formation	Reversible assembly and ATP triggered polymerisation or invader strand	Study nanotube container properties
Demonstrated unilamellar nature of GUV using protein nanopore flux	Location control of fibers along membranes	Vesicle or gold nanoparticle guided transport	DNA-RNA nanotube formation inside compartments
Customisable DNA cytoskeleton properties including chemical functionality (cholesterols) and stiffness	Characterisation of DNA nanotubes and fibers outside of protocells low concentrations	Characterisation of DNA nanotubes and fibers outside of protocells low concentrations	In situ transcription triggered formation of DNA-RNA nanotubes
Formation of cytoskeleton fibers inside protocells via thermal annealing			Enzyme mediated assembly and disassembly of DNA-RNA nanotubes
Multiple cytoskeleton types inside protocells (nanotubes + fibers,			

fibers + fibers, cholesterol fibers + fibers)			
Dynamic real time control of cytoskeleton fibers using magnetic particles			
Exoskeleton nanotubes or fibers around protocells			
Novel mechanism of prototissue assembly			
Prototissues with cross-linking DNA fibers			
Multi-domain prototissues			
Protocell dynamics within prototissues			

Comments:

1) Fig. 3B: Do the authors show somewhere that there are actually still filaments forming along the membrane? To me it looks like its basically covering the whole membrane and that filamentous structures are no longer visible. So maybe they did not stay intact, I don't see and FRAP measurements or z-stacks showing that the DNA stays in one place/that there are really filaments as are shown in their sketch.

We have added an additional CLSM image showing 42 mM fibers with cholesterol tethered along the membrane internal periphery (Figure S24). The image shows clearly defined fiber textures to prove the fibers are still intact once coated along the membranes.

2) The assembly of protocell tissues is the most novel aspect; especially with the two-layer prototissue assembly. It would be great to see them use that and explore it more, e.g. how stiff is the tissue? Can the stiffness be changed? Could one attach filaments to the inside of the GUVs and then form the tissue; does that change how the internal scaffold behaves? Is there a difference in stability between GUVs in the middle of the tissue and the outside? Is the tissue actually intact (so, can it be lifted?)

These are all very interesting questions some of which we have addressed, but others fall outside the remit of this current paper that focuses on establishing the assembly protocol and mechanism or formation.

We show cross-linked prototissues can be moved using a pipettor. The prototissue shears into smaller mobile fragments (Figure S45).

3) The part with the blood cells does not seem to fit in. Why it is important for the rest of the story? Why blood cells in particular? GUVs and cells have been integrated before.

Our research is centred on biomedical applications of synthetic materials. It is therefore very important that our methodologies are compatible to conventional drug administration routes. This means the protocells should be compatible with circulating blood cells. We have modified the introduction to clarify this aspect further.

4) It would be interesting to see if the GUVs could be made to interact with the cells, for example via the external fibres, or since most of the work is on stability and the prototissue, whether the cells interact/can interact or even be incorporated into the prototissue somehow. Other than it feels like this part is from a different story/should only be a side note and not at the end of the story here.

We are indeed interested in exploring these points but, again, feel this is outside the remit of the current publication. As discussed above, this paper is about establishing the bottom-up assembly process of both protocells and prototissues.

5) Small comment: Several figures are not referenced correctly in the main text.

Thank you for pointing these out, we have gone through the text and corrected any mistakes.

REVIEWER COMMENTS

Reviewer #1 (Remarks to the Author):

The authors have addressed all my concerns; I am happy to endorse publication of the revised manuscript.

Reviewer #2 (Remarks to the Author):

The authors have addressed all my comments.

Two clarifications on the revised comparison with the literature, in particular Zhan et al and Agarwal et al - I think they should be mentioned in the intro rather than the discussion; Agarwal et al did show thermal annealing of nanotubes in droplets, in the absence of PEG. Zhan et al did not obtain reversible assembly of nanotubes in droplets (this is mentioned in the rebuttal tables), and their claim of transport is not clearly supported by the data.

Reviewer #3 (Remarks to the Author):

Thank you to the authors for their additional work. I have to say I still have some serious doubts:

Comment 1: The image that the authors added does not confirm that the filaments are intact. Figure S24 looks more like aggregates. I would expect to see FRAP measurements of the lipids and the filaments, which the authors could easily perform given that their data was taken on a CLSM.

Comment 2: The fact that cells don't die when the filaments are added does not mean that they are biocompatible. They may still solicit a pronounced immune response (which would be crucial to address for the tissue use case). The fact that the immunogenicity has NOT been tested should at least be discussed in the manuscript.

Comment 3: In their table, the authors claim that they "Revealed DNA nanotubes and fibers after folding are not in dynamic equilibrium". This is a big claim that lacks experimental proof. What keeps their system out of equilibrium? In my opinion they just see Brownian diffusion in an equilibrated system. This has to be clarified.

Round 2:

Reviewer #1 (Remarks to the Author):

The authors have addressed all my concerns; I am happy to endorse publication of the revised manuscript.

Thank you very much.

Reviewer #2 (Remarks to the Author):

The authors have addressed all my comments.

Two clarifications on the revised comparison with the literature, in particular Zhan et al and Agarwal et al - I think they should be mentioned in the intro rather than the discussion; Agarwal et al did show thermal annealing of nanotubes in droplets, in the absence of PEG. Zhan et al did not obtain reversible assembly of nanotubes in droplets (this is mentioned in the rebuttal tables), and their claim of transport is not clearly supported by the data.

In the revised manuscript, we now also mention the two publications in the introduction section (main text page 3). The table below comparing the existing literature with the new findings of our paper has also been updated.

This paper	Jahnke 2022	Zhan et al 2022	Agarwal 2021
Three lipid formulations with chemistry insight	One lipid composition	Water-in-oil droplets	Water-in oil-droplets
Characterisation of DNA nanotubes and fibers outside of protocells at high concentrations using magnesium	Light triggered reversible assembly of cytoskeletons	Single cytoskeleton type	Nanotube isothermal formation over time with or without PEG
Revealed DNA nanotubes and fibers after folding are not in dynamic equilibrium under our conditions	PEG crowding agents induced fiber formation	Assembly and ATP triggered polymerisation or invader strand	Study nanotube container properties
Demonstrated unilamellar nature of GUVs using protein nanopore flux	Location control of fibers along membranes	Vesicle or gold nanoparticle alignment along cytoskeletons	DNA-RNA nanotube formation inside compartments
Customisable DNA cytoskeleton properties including chemical functionality (cholesterols) and stiffness	Characterisation of DNA nanotubes and fibers outside of protocells at low concentrations	Characterisation of DNA nanotubes and fibers outside of protocells at low concentrations	In situ transcription triggered formation of DNA-RNA nanotubes
Formation of cytoskeleton fibers inside protocells via thermal annealing			Enzyme mediated assembly and disassembly of DNA-RNA nanotubes
Multiple cytoskeleton types inside protocells (nanotubes + fibers, fibers + fibers, cholesterol fibers + fibers)			

Dynamic real time control of cytoskeleton fibers using magnetic particles			
Exoskeleton nanotubes or fibers around protocells			
Novel mechanism of prototissue assembly			
Prototissues with cross-linking DNA fibers			
Multi-domain prototissues			
Protocell dynamics within prototissues			
Cross-linked prototissues can be cleaved from the substrate			

Reviewer #3 (Remarks to the Author):

Thank you to the authors for their additional work. I have to say I still have some serious doubts:

Comment 1: The image that the authors added does not confirm that the filaments are intact. Figure S24 looks more like aggregates. I would expect to see FRAP measurements of the lipids and the filaments, which the authors could easily perform given that their data was taken on a CLSM.

May we politely disagree with this assessment, Figure S24 shows defined fiber textures along the membrane periphery, please see magnified region of interest (Figure S24 A, inset, and page 10 main text). To satisfy the reviewer we have conducted the requested FRAP experiments which further confirm that the cholesterol-labeled fibers remain intact once bound to the membrane interface (please see new Figure S24 B). In these experiments the FRAP profiles of single strand cholesterol (control) and cholesterol- fibers inside GUV containers were collected. FRAP analysis shows the fluorescence of the single strand cholesterol recovered within 150 seconds after the bleaching event, whilst the cholesterol fiber displayed no fluorescence recovery within this timeframe. These results revealed the single strand cholesterol molecules diffuse quickly, comparable to freely diffusing molecules in a lipid bilayer, whilst the cholesterol fibers did not which confirmed their higher order assembly.

Comment 2: The fact that cells don't die when the filaments are added does not mean that they are biocompatible. They may still solicit a pronounced immune response (which would be crucial to address for the tissue use case). The fact that the immunogenicity has NOT been tested should at least be discussed in the manuscript.

Cell viability is a fundamental and established method to demonstrate biocompatibility (<https://pubs.acs.org/doi/10.1021/nn203161y>, <https://pubs.acs.org/doi/10.1021/acsnano.0c07915>, <https://iopscience.iop.org/article/10.1088/1748-605X/abe5fa>). We agree that other parameters, including the immune response to synthetic materials, is important. There are several possible assays to assess immunogenicity. Previously we assessed the pro-inflammatory cytokine release (TNF-alpha) from healthy volunteer (human) immune cells after addition of the protocells, GUVs, DNA nanotubes, and DNA fibers at 6 h (Figure S47). To satisfy the referee, we have now expanded our immunological studies by monitoring interleukin-6 (IL-6) response. The updated Figure 47 shows low doses (1 μ M) of single stranded DNA, DNA fibers and GUVs elicited minimal increase in IL-6, whereas the same concentration of DNA nanotubes was associated with a significant increase in IL-6. A similar trend was seen with TNF-alpha. Levels of proinflammatory cytokines were higher at DNA concentrations of 10 μ M. The viability results show the protocells and DNA macrostructures display promising biocompatible

properties to RBCs and WBCs under the assayed conditions. However, additional work is required to minimise their immunologic responses, for example, by introducing PEG groups in lipids or by coating the DNA macrostructures in serum proteins. We have now edited the text accordingly in the results section (main text page 19-20).

Comment 3: In their table, the authors claim that they "Revealed DNA nanotubes and fibers after folding are not in dynamic equilibrium". This is a big claim that lacks experimental proof. What keeps their system out of equilibrium? In my opinion they just see Brownian diffusion in an equilibrated system. This has to be clarified.

Our data shows that DNA nanotubes and fibers are not in dynamic equilibrium after folding under our conditions, either inside protocells or in solution. As discussed in the original response to Reviewers 1 and 2 (please see below), we monitored the dynamic rearrangement of NT and F, or F and F combinations outside protocells after 1 min and 48 hours (Figure S23). The results showed that neither NT nor F in any combination dynamically rearranged. If the constructs were in dynamic equilibrium then the dye signals of NT and F would merge over the course of 48 hours. However, we did not observe any dye fluorescence merging thus confirming they are not in dynamic equilibrium under our conditions.

REVIEWERS' COMMENTS

Reviewer #3 (Remarks to the Author):

I have no further comments, thank you.